# Integrated evolutionary analysis reveals antimicrobial peptides with limited resistance

Réka Spohn[1,10], Lejla Daruka[1,2,10], Viktória Lázár[1,8], Ana Martins[1], Fanni Vidovics[1], Gábor Grézal [1,3], Orsolya Méhi[1], Bálint Kintses[1,4], Mónika Számel[1,2], Pramod K. Jangir[1,2], Bálint Csörgő[1,9], Ádám Györkei[1,3], Zoltán Bódi[1], Anikó Faragó[2,4], László Bodai [4], Imre Földesi[5], Diána Kata[5], Gergely Maróti[6], Bernadett Pap[6], Roland Wirth[7], Balázs Papp[1,3] & Csaba Pál [1]*

Antimicrobial peptides (AMPs) are promising antimicrobials, however, the potential of bacterial resistance is a major concern. Here we systematically study the evolution of resistance to 14 chemically diverse AMPs and 12 antibiotics in *Escherichia coli*. Our work indicates that evolution of resistance against certain AMPs, such as tachyplesin II and cecropin P1, is limited. Resistance level provided by point mutations and gene amplification is very low and antibiotic-resistant bacteria display no cross-resistance to these AMPs. Moreover, genomic fragments derived from a wide range of soil bacteria confer no detectable resistance against these AMPs when introduced into native host bacteria on plasmids. We have found that simple physicochemical features dictate bacterial propensity to evolve resistance against AMPs. Our work could serve as a promising source for the development of new AMP-based therapeutics less prone to resistance, a feature necessary to avoid any possible interference with our innate immune system.

[1] Synthetic and Systems Biology Unit, Institute of Biochemistry, Biological Research Centre, Szeged, Hungary. [2] Doctoral School of Biology, Faculty of Science and Informatics, University of Szeged, Szeged, Hungary. [3] HCEMM-BRC Metabolic Systems Biology Lab, Szeged, Hungary. [4] Department of Biochemistry and Molecular Biology, University of Szeged, Szeged, Hungary. [5] Department of Laboratory Medicine, University of Szeged, Szeged, Hungary. [6] Institute of Plant Biology, Biological Research Centre, Hungarian Academy of Sciences, Szeged, Hungary. [7] Department of Biotechnology, University of Szeged, Szeged, Hungary. [8] Present address: Faculty of Biology, Technion - Israel Institute of Technology, Haifa, Israel. [9] Present address: University of California, San Francisco, Department of Microbiology and Immunology, San Francisco, CA, USA. [10] The authors contributed equally: Réka Spohn, Lejla Daruka. *email: cpal@brc.hu

Antimicrobial peptides (AMPs) are short, structurally diverse peptides with a broad spectrum of antibacterial activities. Although there is considerable diversity in the amino acid content, length, and structure among peptides, typically, they are positively charged and amphipathic[1]. AMPs are present in all classes of life, and are inherent part of the defense mechanisms against microbial pathogens. In vertebrates, many provide protection against pathogens and also have modulatory roles in the innate immune system[2,3]. AMPs frequently target the bacterial membrane by forming pores on it. This process is initiated by the interaction between the positively charged AMP and the negatively charged components of bacterial cell wall, including lipopolysaccharides (LPS) and phospholipids[4]. AMPs can also interfere with central cellular processes, such as DNA and protein syntheses, protein folding, and cell wall synthesis[5,6]. Given their broad spectrum of activity, much effort has been made to find potential novel antibacterial drug candidates among AMPs, although skeptics have raised issues concerning low peptide stability, costly production, and pleiotropic biological effects[7,8].

Evolution of resistance against AMPs was initially thought to be less likely due to the rapid bactericidal effects and multiple potential bacterial targets of these compounds[1,9]. However, small-scale studies indicate several major mechanisms of resistance, including the bacterial release of molecules that neutralize the exogenous peptide, enzymatic decay by bacterial proteases, and the modification of the outer bacterial membrane to prevent AMP binding or uptake[10–14]. Also, there are reasonable concerns that the therapeutic use of certain AMPs could drive a rapid evolution of resistance to host-defense peptides with immune-related functions in the human body[15,16]. This is a pressing issue, as several synthetic AMPs are currently tested in clinical trials (the majority of them are proposed for topical use)[2,3].

However, the scope of AMPs is broad. There are thousands of registered AMPs derived from a wide range of organisms, with distinctive physicochemical, structural and functional characteristics[17,18]. Therefore, major variations may exist in the probability or resistance of different AMPs. There is an increasing demand for a better understanding of AMP-specific resistance mechanisms, as it would greatly influence the capacity to design more efficient AMP-based drug candidates[2,3,13].

A better understanding of AMP-specific resistance demands studying multiple AMPs in a range of bacterial species. Moreover, bacteria can acquire resistance through diverse genetic mechanisms, including point mutations, amplification of genomic segments, and horizontal transfer of resistance genes. Therefore, these mechanisms should be studied systematically in the laboratory. Finally, it is of central importance to compare the rate of resistance evolution against AMPs and clinically employed antibiotics.

Here we systematically characterize bacterial potential to acquire resistance against a set of chemically diverse AMPs (Supplementary Table 1). For this purpose, we combine laboratory evolution, systematic gene overexpression studies and functional metagenomics. We integrate the results of these complementary tests to get a global overview of AMP resistance evolution. We reveal a limited probability of resistance development against certain AMPs. Such AMPs could serve as a promising source for the development of resistance-free, AMP-based drug candidates.

## Results

**Laboratory evolution of resistance to AMPs.** Our first goal was to study resistance evolution against a set of AMPs and antibiotics. We have followed established protocols with minor modifications to evolve bacterial populations under controlled laboratory conditions[19,20]. The applied protocol aims to maximize the level of drug resistance in the evolving populations achieved during a fixed time period. We have initiated a laboratory evolution experiment, starting with a single clone of *Escherichia coli* K-12 BW25113 (abbreviated as *E. coli* K-12 BW25113). Parallel evolving populations were exposed to increasing concentrations of one of 14 AMPs (Supplementary Table 1). Peptides were chosen based on the following criteria: diverse sources (synthetic/natural), different putative mechanisms of action, structural diversity, and clinical relevance. Another set of populations were exposed to one of 12 clinically relevant antibiotics (Supplementary Table 2), using the same protocol. Accordingly, we have propagated 10 independent populations in the presence of each AMP or antibiotic for ∼120 generations. We isolated a single, representative clone from each population, resulting in 140 AMP-adapted lines and 120 antibiotic-adapted lines. Throughout the paper, we compare the level of resistance in the evolved lines to the corresponding wild-type strain by measuring changes in the drugs' minimum inhibitory concentrations (MIC).

In agreement with prior studies, the evolution of resistance against small-molecule antibiotics was a rapid process[21–23]. The median antibiotic resistance levels were ∼90 times higher compared to the wild-type strain (Fig. 1a and Supplementary Data 1). Resistance levels induced by AMP treatments were significantly lower and more heterogeneous on average than the resistance levels evoked by antibiotic treatments (Fig. 1a). Like most antibiotics studied here, certain AMPs are also prone to resistance. For instance, polymyxin B (PXB)—an AMP used as a last-resort drug in the treatment of multidrug-resistant Gram-negative bacterial infections[24]—is a notable example. During the course of our laboratory evolution, bacterial susceptibility to PXB decreased, reaching resistance levels of over 3000-fold higher than that of the wild-type strain (Fig. 1a, Supplementary Data 1). In agreement with other laboratory studies, we have also isolated resistant mutants from populations exposed to pexiganan (PEX), protamine (PROA), and PR-39 stresses[25–27].

However, there was a low probability of resistance development against some other AMPs. In particular, the average increment in resistance level was less than threefold in populations exposed to cecropin P1 (CP1), indolicidin (IND), peptide glycine-leucine amide (PGLA) and pleurocidin (PLEU) stresses (Fig. 1b). Strikingly, we have failed to detect any resistance in the evolved lines exposed to tachyplesin II (TPII) and R8 (Fig. 1a, Supplementary Data 1). However, bacterial lines with an elevated mutation rate (mutators) are known to display an exceptionally rapid evolution of antibiotic resistance[28]. Therefore, we have tested whether resistance against TPII would emerge in such mutator populations, but none of the 10 laboratory evolved mutator *E. coli* K-12 mutD5 populations displayed resistance to TPII (Fig. 1a, Supplementary Data 1).

Next, we have also studied laboratory evolution under TPII and PXB stresses in a range of pathogenic bacteria to see whether the limited AMP resistance observed in *E.coli* K-12 BW25113 holds in other bacterial strains and species. The tested pathogens initially sensitive to the AMPs and the antibiotics included *Escherichia coli* (ATCC® 25922™) (abbreviated as *E. coli* ATCC 25922), *Salmonella enterica* subsp. *enterica* serovar Typhimurium LT2 (abbreviated as *S. enterica* LT2), *Klebsiella pneumoniae* subsp. *pneumoniae* (Schroeter) Trevisan (ATCC® 10031™) (abbreviated as *K. pneumoniae* ATCC 10031) and *Acinetobacter baumannii* Bouvet and Grimont (ATCC® 17978™) (abbreviated as *A. baumannii* ATCC 17978). In agreement with the results seen with *E. coli* K-12 BW25113, a high level of PXB-resistance has emerged in all pathogens studied: certain lines have displayed an

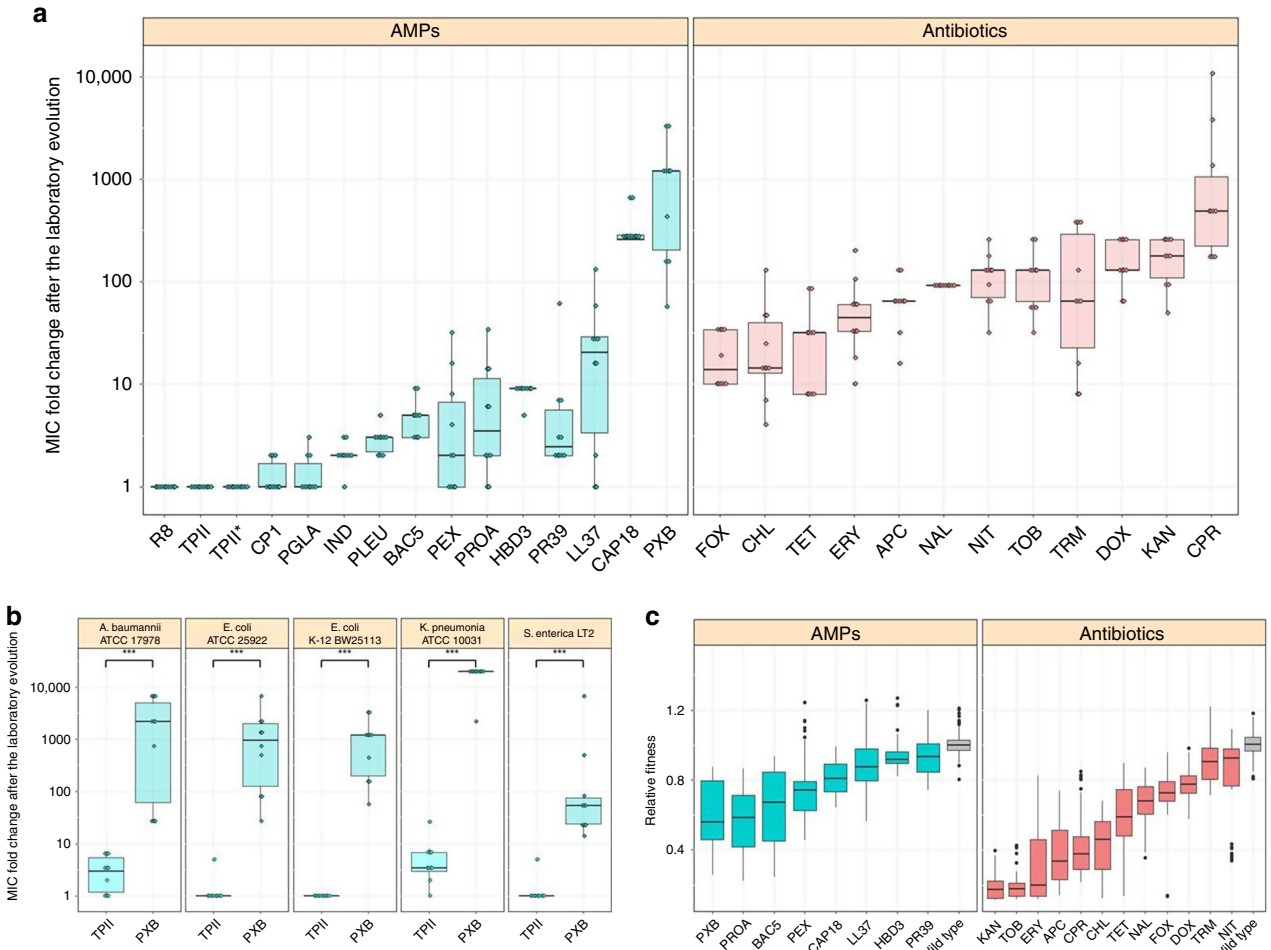

**Fig. 1** MIC and relative fitness of adapted lines after the laboratory evolution. **a** Relative resistance level in laboratory evolved *E. coli* K-12 BW2511 lines exposed to one of each 14 AMPs (blue) or 12 antibiotics (red), respectively (at least 9 parallel-evolved lines per drug). Altogether, lines exposed to AMPs ($N = 138$) developed significantly lower resistance, than lines exposed to antibiotics ($N = 120$) ($P < 0.0001$, one-sided permutation test). The resistance levels reached were more heterogeneous across AMP treatments ($N = 14$) than across antibiotic treatments ($N = 12$) ($P = 0.03478$ *F*-test). Each data point represents the MIC fold change of one of each parallel-evolved line. The mutD5 mutator strain exposed to TPII is marked by an asterisk (*). **b** Relative resistance level after laboratory evolution in clinical isolates under TPII or PXB stresses, respectively. Evolved lines exposed to TPII reached significantly lower resistance level than lines exposed to PXB (*** indicate the significant difference at least *P*-value = $1.65 \times 10^{-4}$, two-sided Mann–Whitney test, $N = 10$ each group). Each data point represents the MIC fold change of one of each parallel-evolved line. **c** Relative fitness of 60 antibiotic-resistant and 38 AMP-resistant lines displaying at least twofold increments in resistance level to the drug indicated. Fitness was measured as the area under the growth curve in an antibacterial agent-free medium and was normalized to that of the wild-type (gray color). Throughout Fig. 1, boxplots show the median, first and third quartiles, with whiskers showing the 5th and 95th percentile. For AMP and antibiotic abbreviations, see Supplementary Tables 1–2. Data in this figure are representative of at least two biological replicates. Source data are provided as a Source Data file

increment of resistance level as high as 20,000 times compared to the corresponding wild-type strain (Fig. 1b, Supplementary Data 1). By sharp contrast, resistance to TPII has remained at very low levels in all species (Fig. 1b, Supplementary Data 1). Further studies should explore potential species-specific differences in resistance development.

**Fitness costs of resistance.** Prior studies have demonstrated that antibiotic resistance frequently incurs a moderate fitness cost in the absence of drug, analyzed by measuring growth rates in laboratory conditions[29]. Such fitness costs seem to influence the spread of antibiotic-resistant bacteria in clinical settings[30]. However, fitness data on AMP-adapted lines is far more limited[31].

In this analysis, we focused on 60 antibiotic-adapted, 38 AMP-adapted lines, all of which displayed at least twofold increment in resistance level to the drug they had been exposed to during the

course of laboratory evolution. We estimated fitness by measuring growth of these strains and the corresponding wild-type in antibiotic-free medium ("Methods"). In total, 96.7% of the antibiotic-resistant lines showed significantly reduced growth compared to the wild-type strain, while this figure is 92.1% in the case of AMP-adapted lines (Fig. 1c, Supplementary Data 2). We conclude that the frequency of AMP ($N = 38$) and antibiotic ($N = 60$) resistant strains with fitness cost are comparable to each other ($P = 0.37$, Fisher's two-sided exact test). As might be expected[13], fitness costs varied substantially across antibiotics and AMPs (Fig. 1b). Intriguingly, however, the fitness cost of AMP-adapted lines ($N = 38$) was significantly lower than that of antibiotic-adapted lines ($N = 60$) ($P < 2.2e-16$, two-sided Mann–Whitney *U* test). This pattern is unlikely to reflect differences in the level of resistance between AMP and antibiotic strains, respectively. There is only a weak, marginally significant negative correlation between the level of resistance and fitness

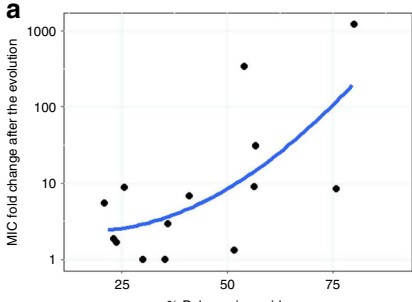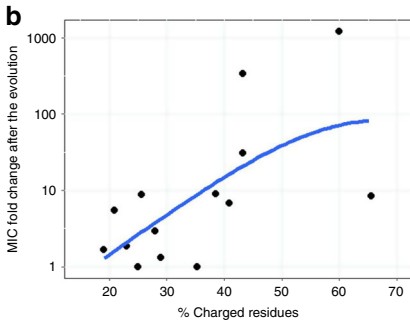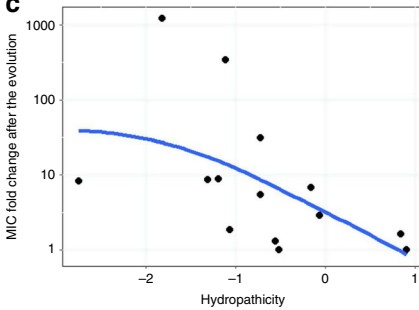

**Fig. 2** Resistance level correlate with AMPs' physicochemical features. Each datapoint shows the average MIC-fold change in laboratory evolved *E. coli* K-12 BW2511 lines exposed to one of each 14 AMPs. **a** Fraction of polar amino acids and relative resistance level (Spearman's rho = 0.58; $p = 0.03$, $N = 14$). **b** Fraction of positively charged and relative resistance level (Spearman's rho = 0.62; $p = 0.02$, $N = 14$). **c** AMP hydropathicity and relative resistance level (Spearman's rho = $-0.73$; $p = 0.002$, $N = 14$). For AMP properties, see Supplementary Data 3. Blue lines indicate the curve fitted using LOESS smoothing method in R. Source data are provided as a Source Data file

cost (Supplementary Fig. 1, Spearman's rho = $-0.24$, $P = 0.016$ ($N = 98$)), and this correlation becomes non-significant after controlling for drug type ($P = 0.6$ and $P = 0.39$ for antibiotics ($N = 60$) and AMPs ($N = 38$) respectively, Spearman's correlation). Together, these results indicate that the lack of resistance against certain AMPs during laboratory evolution cannot be explained by differences in fitness costs.

**Physicochemical features of AMPs with limited resistance.** Next, we have investigated how amino acid physicochemical properties influence the rate of AMPs resistance evolution. To this end, we have calculated the fraction of charged, polar and hydrophobic amino acids for each AMP (Supplementary Data 3), and have correlated those features with the average increment in resistance level achieved during laboratory evolution (Fig. 1a). The analysis has revealed that the AMPs less prone to resistance tend to contain fewer polar and positively charged amino acids, and display increased hydropathicity (Fig. 2).

**No cross-resistance to AMPs with limited resistance.** Next, we have explored whether laboratory evolved AMP resistance provide any cross-resistance to AMPs which appear to be less prone to resistance. For this purpose, we have isolated 38 independently evolved lines, all of which have displayed resistance to the AMP they had been exposed to in the laboratory (at least four lines per AMP, see Supplementary Data 1). We have measured the changes in susceptibilities of these strains to the AMPs (PLEU, PGLA, R8, IND, TPII, and CP1) towards which resistance evolution was limited, as well as to the human peptide LL37. LL37 is a cathelicidin-derived AMP present throughout the human body with critical defensive roles[32]. Cross-resistance was defined as a minimum of twofold increase in resistance level.

Three main findings have emerged from the analysis. First, 13.4% of all possible combinations of the AMPs and the evolved lines have displayed cross-resistance (Fig. 3, Supplementary Data 4). Second, only 5 out of the 33 non-LL37 adapted lines have shown cross-resistance to the human peptide LL37. Moreover, 4 out of these 5 adapted lines had been exposed to CAP18 during the course of laboratory evolution. As CAP18 belongs to the cathelicidin family[32], chemical relatedness may drive cross-resistance to LL37. Third, and most significantly, none of the 38 AMP-resistant lines displayed cross-resistance to R8, TPII, and CP1 (Fig. 3, Supplementary Data 4). These results further confirm that resistance mutations against these specific AMPs are particularly rare.

**Molecular mechanisms underlying AMP resistance.** To identify mutational events underlying AMP resistance, the 38 AMP-adapted lines characterized above were subjected to whole-genome sequencing using Illumina platform. We have implemented an established computational pipeline to identify point mutations, small insertions and deletions that had accumulated during the course of laboratory evolution[21,22]. Altogether, 197 independent mutational events have been identified (Supplementary Data 5). On average, we have detected 1.45 deletions, 0.61 insertions and 2.21 single nucleotide polymorphisms (SNPs) per a single clone. To statistically test whether the ratio of non-synonymous to synonymous SNPs was higher than expected based on a neutral model of evolution, we employed an established method[33], that is especially well-suited for experimental evolution studies with limited number of observed mutations[34]. Briefly, we took all different point mutations observed in protein coding regions and calculated the probability that 94% or more substitutions would result in a non-synonymous substitution if it occurred in a random coding position. The excess of non-synonymous substitution observed in the evolved genomes was significant ($p = 0.000004$, one-sided permutation test[34]).

Thirty percent of the affected genes were mutated repeatedly (i.e., at least three times), and 94% of these genes have acquired mutations in lines adapted to at least two different AMPs (Figs. 4a, b, Supplementary Data 5). Several repeatedly mutated proteins and molecular pathways had been previously linked to AMP resistance in laboratory or clinical settings (Supplementary Data 6). The most relevant proteins were members of the waa-operon and the Mla-pathway involved in LPS and phospholipid trafficking, respectively, and the BasS-BasR two-component sensor system. This two-component system regulates both the LPS pathway and phospholipid trafficking[35].

Next, we have investigated whether mutations in the most relevant genes provide resistance to those AMPs which are less prone to resistance (e.g., TPII, R8 and CP1). Thus, a selected human beta-defensin-3 (HBD3) resistant line carrying a single mutation in BasS (*basS* L194R), mlaD (*mlaD* a342del) and WaaY (*waaY* M1†) was chosen for further analysis. The observed mutations were inserted individually into the *E. coli* K-12 BW25113. The mutations conferred resistance to human HBD3, but no resistance to R8, CP1 or TPII (Supplementary Data 4). We conclude that commonly observed AMP resistance mutations provide no resistance to these AMPs.

**AMP resistance via reduced net negative surface charge.** LPS are major components of bacterial outer membrane, responsible for regulating permeability and contributing to negative surface

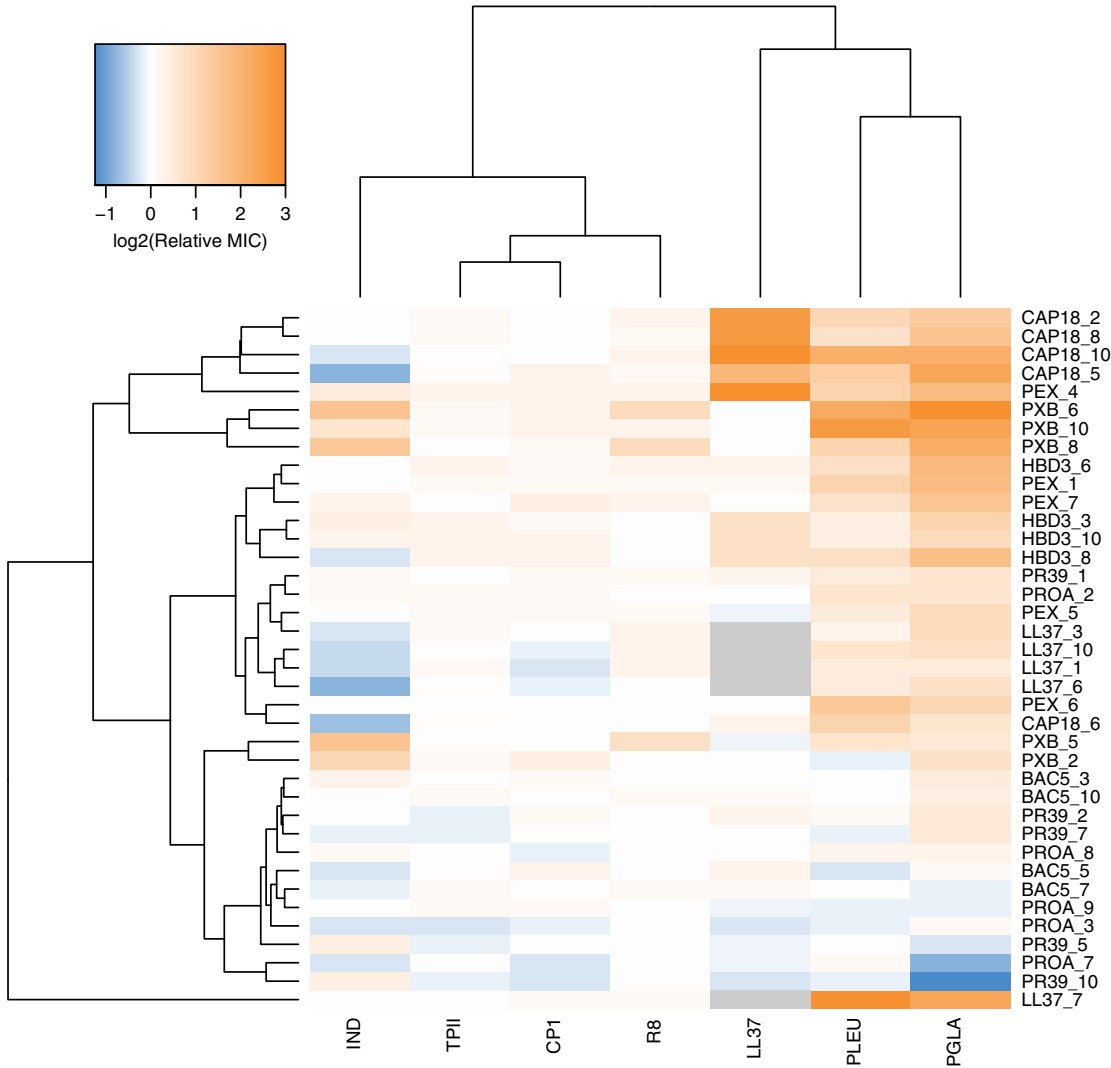

**Fig. 3** Cross-resistance of AMP-resistant lines towards a set of 7 AMPs. Relative minimum inhibitory concentration (MIC) was calculated as the ratio of the MIC of the resistant line and the sensitive wild-type strain. Hierarchical clustering was performed separately on rows and columns, using complete linkage method with Euclidean distance measure on the raw MIC data. Throughout the figure, blue coloring refers to collateral-sensitivity (MIC at least two-fold lower than the wild-type), orange coloring refers to cross-resistance (MIC at least two-fold higher than the wild-type), white coloring refers to no or minimal change in susceptibility (MIC in between). Gray coloring refers to not applicable. AMP-resistant lines (rows) are named after the AMP it has been exposed to during the evolutionary experiment, and the serial number of the corresponding population. For relative MICs, see Supplementary Data 4 and for AMP abbreviations, see Supplementary Table 1. Data in this figure are representative of at least two biological replicates

charge[36]. Consequently, LPS composition has a major impact on bactericidal efficiency of cationic AMPs. As proteins related to several mutated genes regulate outer membrane or LPS composition in particular (Supplementary Data 5), we hypothesized that bacterial populations developed AMP resistance partly via the modifications of LPS structure, ultimately leading to a reduced net negative surface charge of the outer membrane. To investigate this hypothesis, we have carried out surface charge measurements on 16 representative AMP-adapted lines (two lines per AMP). The analysis has revealed that 13 of the 16 AMP-adapted lines tested displayed significantly reduced negative surface charge of the outer bacterial membrane compared to the wild-type strain (Fig. 5). These lines were frequently found to carry mutations in the BasR-BasS two-component sensor system. Indeed, the representative mutation in BasS, when introduced into *E. coli* K-12 BW25113, caused a major reduction in net negative surface charge, potentially explaining its impact on resistance (Fig. 5). However, it is probable that many other mutated genes contribute

to the observed changes in negative surface charge of the outer membrane.

**The impact of gene amplification on resistance**. Bacterial gene amplification generates extensive genetic variation that can provide drug resistance[37,38]. As gene amplification events tend to be genetically unstable and reduce bacterial fitness, they are transient in nature[39]. Using the ASKA library, which comprises each *E. coli* ORF cloned into an expression vector[40], we have conducted a global survey to evaluate whether artificially amplified genes provide resistance to any of the 14 AMPs and 11 antibiotics tested. As chloramphenicol (CHL) was used as a selection marker for the ASKA plasmid[40], this antibiotic was not tested in this screen. *E. coli* K-12 BW25113 cells were transformed with the pooled plasmids of the ASKA library, and the genetically mixed populations were exposed to increasing levels of AMP or antibiotic stresses, respectively. In all cases, the employed AMP or antibiotic dosages were above the MIC of the wild type. For each

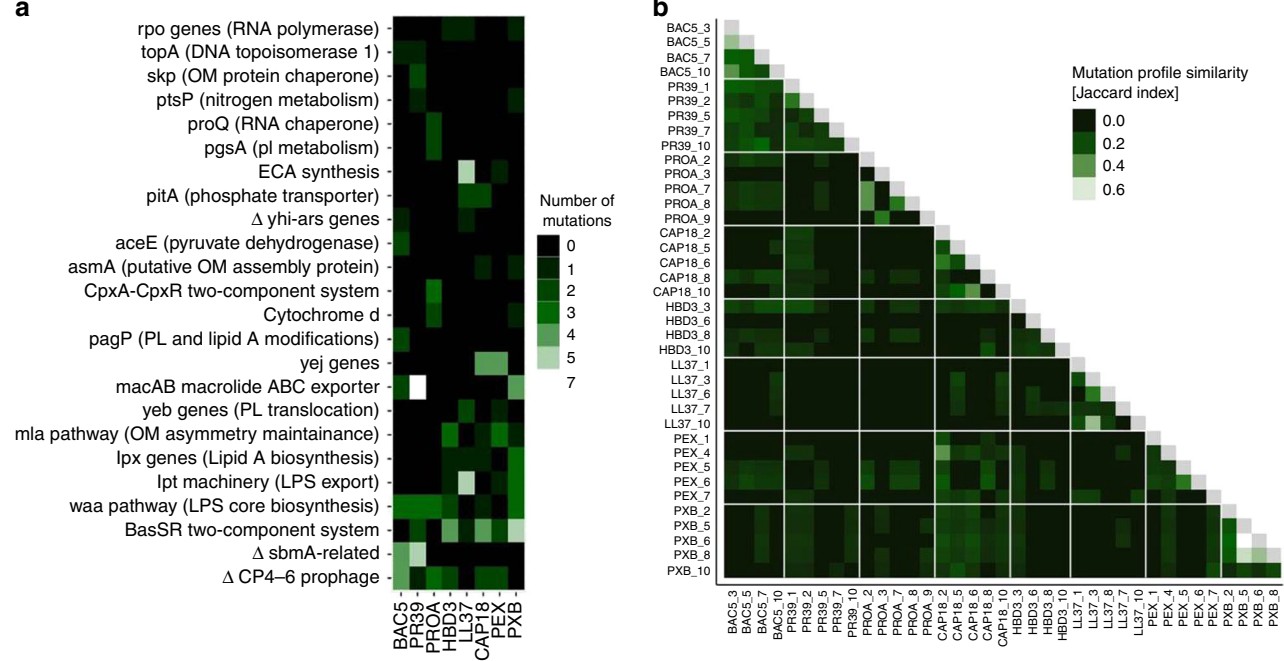

**Fig. 4** Mutational profiles of 38 AMP-resistant lines. **a** The figure shows cellular complexes and pathways mutated independently in multiple lines as a function of AMPs used during laboratory evolution. The color code indicates the number of individual mutations affecting a given cellular subsystem in lines evolved to a given AMP. OM outer membrane, PL phospholipid, LPS lipopolysaccharide, Δ gene deletion. **b** Heatmap shows mutation profile similarity of AMP-resistant lines. Mutation profile similarity between each pair of AMP-resistant lines was estimated by the Jaccard's coefficient between their set of mutated genes. Large deletions were counted as one gene. AMP-resistant lines are named after the AMP it has been exposed to during the evolutionary experiment, and the serial number of the corresponding population. For AMP abbreviations, see Supplementary Table 1

drug, we identified the MIC provided in the pooled ASKA plasmid library, and compared it to the MIC of the wild-type strain carrying the empty ASKA plasmid.

When transformed into *E. coli* K-12 BW25113, the pooled ASKA library provides a 2- to 64-fold increment in antibiotic resistance level, suggesting that gene amplification can have an important contribution to resistance (Fig. 6a, Supplementary Table 3). Consistent with a prior study[41], trimethoprim (TRM) resistance level was found to get increased by 32-fold as a result of gene overexpression. By sharp contrast, the same library induced no detectable increment in resistance level to 8 out of the 14 AMPs tested, including R8, IND, TPII and CP1. Moreover, the artificial gene amplification of native genes was found to provide only a maximum 1.6-fold change in resistance level to the remaining 6 AMPs (Fig. 6a).

**Functional metagenomics of a soil library.** As AMPs are produced by a diverse set of organisms to promote immune defenses, nutrient acquisition or elimination of competitors from the environment, they are found in a many environments[42], including mammalian tissues, soil[43–45] and aquatic environments. For example, *Bacillus polymyxa* inhabits soil, plant roots, and marine sediments[44], and produces clinically relevant polymyxins. Soil is also an ancient reservoir of antibiotic resistance genes and it has been shown that antibiotic resistance genes can be exchanged between soil dwelling and pathogenic bacteria[46]. For all these reasons, we consider soil as a relevant source environment to study the mobilization potential of AMP-resistance genes.

Briefly, metagenomic DNA was cut, and fragments between 1.5–5 kb were shotgun cloned into a plasmid to express the genetic information in *E. coli* K-12 BW25113 (for details see "Methods"). About 1.8 million members of the constructed library, corresponding to a total coverage of 3.6 Gb (the size of ~900 bacterial genomes), were then selected on solid culture medium in the presence of one AMP or antibiotic. We have focused on four AMP candidates found to be less prone to resistance induced by genomic mutations (R8, IND, TPII and CP1, Fig. 1a), and four, clinically relevant small-molecule antibiotics with diverse modes of action (TRM, TOB, CHL, and CPR). Resistant clones were isolated, and the number of unique DNA fragments conferring resistance (i.e., resistance contigs) was determined by sequencing. In agreement with prior studies[46,47], multiple genetically independent resistant clones were found to emerge against all tested antibiotics (Fig. 6b, Supplementary Data 7). By contrast, no resistant clone was observed against the tested AMPs (Fig. 6b). These results suggest that mobile resistance genes against these AMPs are relatively rare in the soil.

**Antibacterial activity and toxicity analysis of TPII and CP1.** As described above, we have identified AMPs that appear to be less prone to mutation- and plasmid-mediated resistance development alike. Two of these AMPs, TPII and CP1, could be promising candidates for future therapeutic considerations. Both CP1 and TPII exhibit broad-spectrum activity against Gram-negative bacteria (Supplementary Table 4). A hemolysis assay indicates that both TPII and CP1 exhibit only a minimal haemolytic effect (<10% hemolysis) even at concentrations two orders of magnitude higher than their MICs against bacteria commonly associated with resistance (i.e., ESKAPE pathogens, see Supplementary Table 5). We calculated the activity/toxicity index (ATI)[48], the ratio of the concentration causing 10% hemolysis (minimum haemolytic concentration, MHC) and the median of MICs (MM). It was found that ATI is relatively high for CP1 and TPII against all bacteria tested (Supplementary Table 6). Future in vivo studies should be performed to confirm these findings.

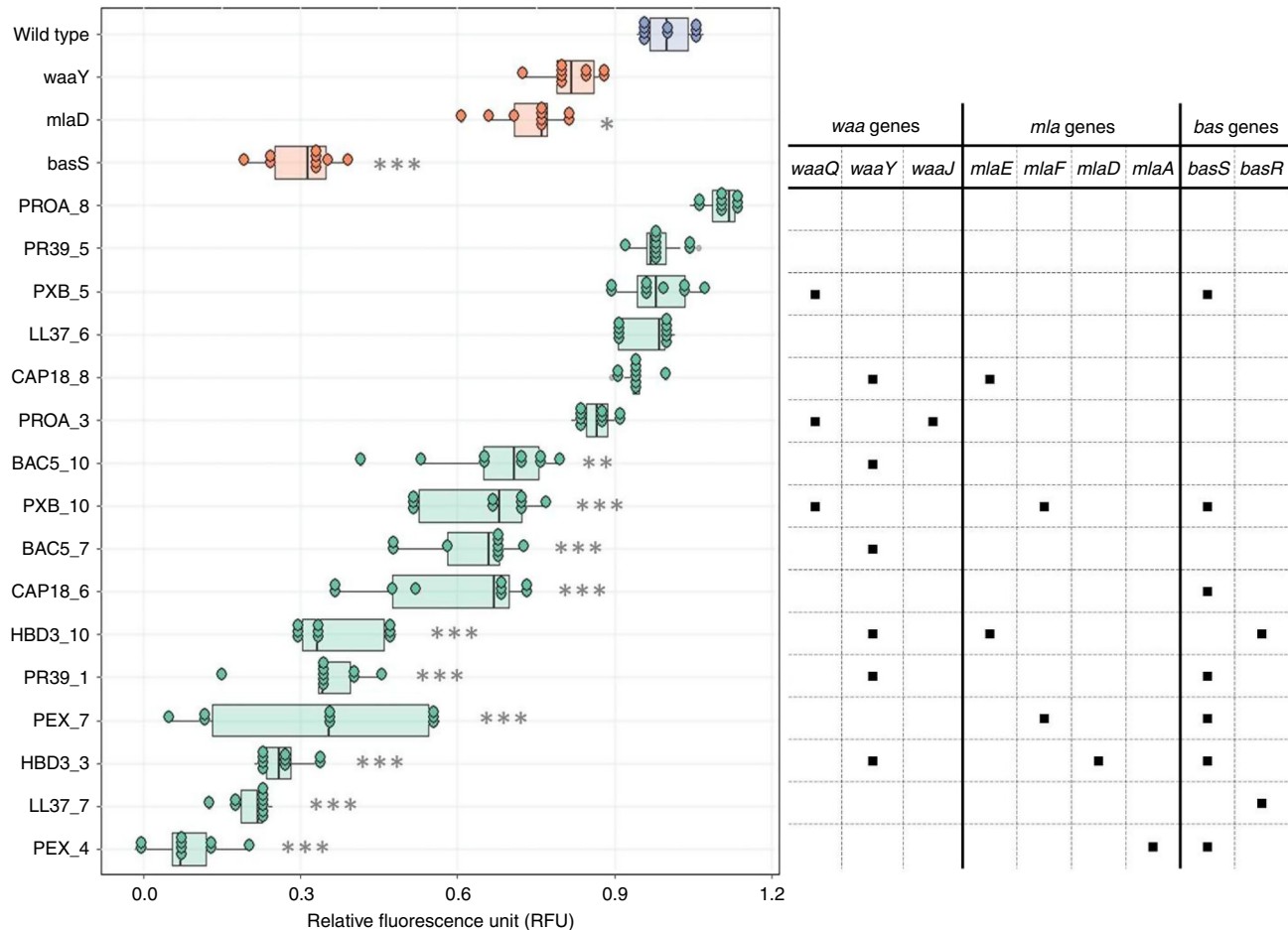

**Fig. 5** Surface charge measurement. Relative surface charge was measured as the relative change in the binding of the positively charged FITC-PLL compared to the wild-type (see Methods). The table on the right side shows whether the adapted line carries a mutation in the *waa* or *mla* pathway genes or in the BasSR two-component system. AMP-resistant lines are named after the AMP it has been exposed to during the evolutionary experiment, and the serial number of the corresponding population. Each data point represents the relative fluorescence unit of one of 9 biological replicate. Boxplots show the median, first and third quartiles, with whiskers showing the 5th and 95th percentile. Significant differences compared to the wild type are marked with gray asterisks (*$P < 0.05$, **$P < 0.01$, ***$P < 0.001$, two-sided Dunnett's test, $N = 9$ each group). Source data are provided as a Source Data file

## Discussion

AMPs are promising novel antimicrobial agents[2], but evolution of resistance against AMPs is a major health concern[14,16]. This issue is all the more important as knowledge on AMP resistance could direct the design of novel AMPs for clinical applications[49]. Predicting the risk of resistance formation against new antibiotic agents is a notoriously difficult task, as a complex interplay of several factors shapes the rise and evolutionary fate of drug resistant bacteria in clinical settings[30]. The most important factors include (a) the frequency by which resistance arises by endogenous mutations or horizontal gene transfer, (b) the level of resistance conferred by combinations of resistance mechanisms, (c) the potential fitness costs of resistance, and (d) the extent of cross-resistance between drugs. In this work, we investigated these issues with a range of clinically relevant antibiotics and functionally/chemically diverse AMPs. Six main patterns have emerged from our study.

First, parallel evolving populations were exposed to increasing concentrations of one of 14 AMPs and 12 clinically relevant antibiotics, respectively. All the antibiotics studied, as well as certain AMPs were found to be prone to resistance (Fig. 1a). For example, susceptibility to PXB substantially decreased during the course of our laboratory evolution, reaching a resistance level of over 3000-fold higher compared to the wild-type strain. This is

important, as PXB and related AMPs are used as last-resort therapeutics in the treatment of multidrug-resistant Gram-negative bacterial infections[50]. We have also identified repeatedly mutated proteins, putatively linked to AMP resistance, including proteins involved in LPS modification and transport (*waa*-operon), phospholipid trafficking (Mla-pathway), and a two-component sensor system (BasR-BasS). These proteins most likely act via the modification of bacterial outer membrane components[19], ultimately leading to a reduced net negative surface charge of the outer membrane (Fig. 5). For other molecular mechanisms associated to AMP resistance, see Supplementary Data 5.

Second, laboratory evolution has revealed that there was a very low probability of resistance against some other AMPs, including CP1, IND, TPII, and R8 (Fig. 1a). The rate by which resistance mutations arise depends on the mutational supply rate (i.e., the product of genomic mutation rate and population size). Intriguingly, even *E. coli* populations with elevated genomic mutation rate failed to develop resistance against TPII. Importantly, AMP-adapted lines were found to display no cross-resistance to R8, TPII, and CP1, indicating that resistance mutations against these AMPs are particularly rare (Fig. 3). We wish to emphasize that the level of resistance in bacterial populations was measured by growth inhibition assays. Therefore, it is an open issue whether

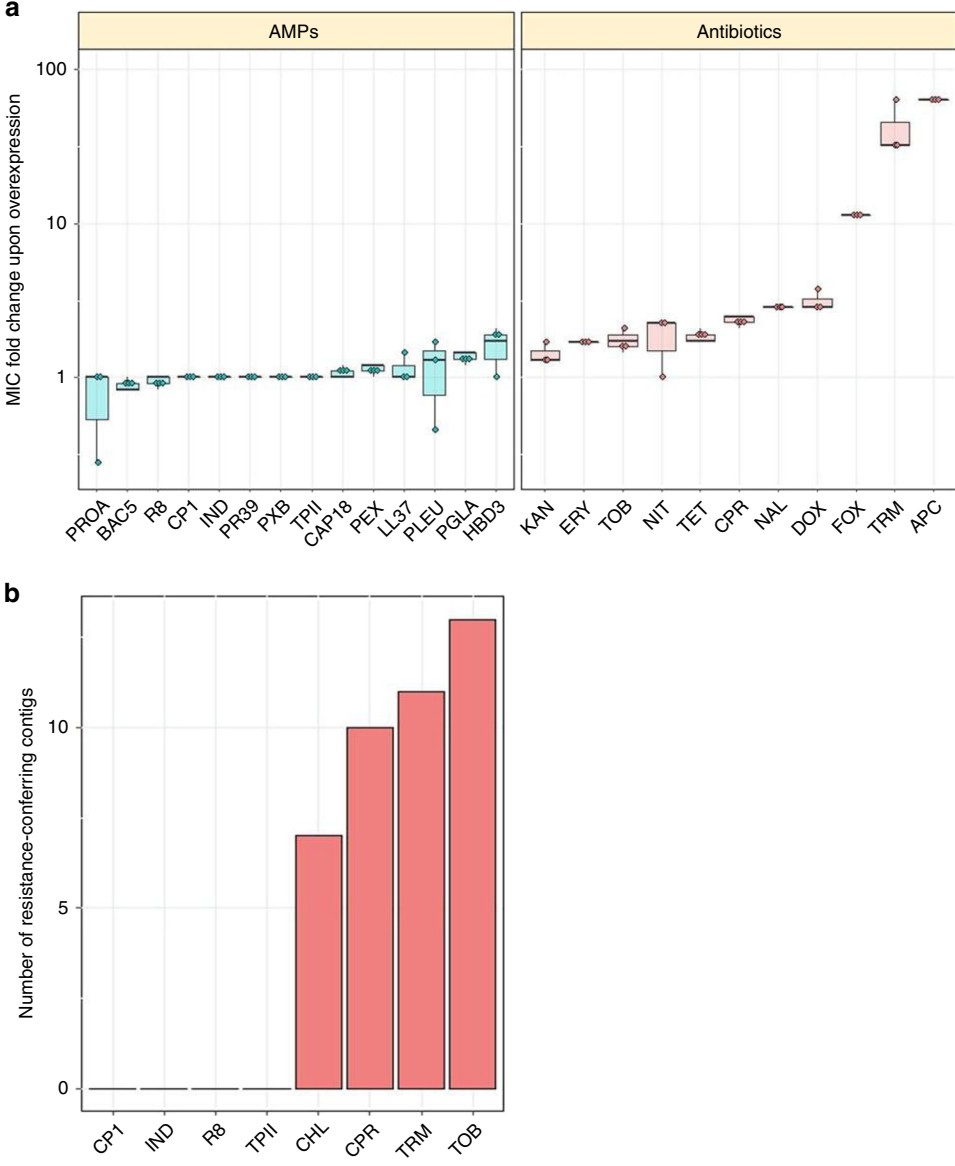

**Fig. 6** Impact of gene amplification and foreign genes on resistance. **a** The impact of gene overexpression on resistance level. Dots represent show the MIC provided by the ASKA plasmid library relative to the MIC of the wild type carrying the empty ASKA plasmid (three biological replicates each). Altogether, the overexpression of the ASKA plasmid library resulted in significantly higher resistance to antibiotics ($N = 11$) than for AMPs ($N = 14$) ($P < 0.0001$, one-sided permutation test). Each data point represents the MIC fold change of one of three biological replicate. Boxplots show the median, first and third quartiles, with whiskers showing the 5th and 95th percentile. **b** Functional metagenomics of a soil library. Functional selection has revealed 41 distinct antibiotic resistance-conferring DNA contigs (red bars), while no AMP resistance-conferring contigs were identified ($P = 2 \times 10^{-16}$ from two-sided negative binomial regression). For AMP and antibiotic abbreviations, see Supplementary Tables 1–2. Source data are provided as a Source Data file

mutations that shape persistence during lethal drug exposure without any major effect on resistance are wide-spread.

Third, the fitness cost is another potentially relevant parameter shaping the long-term fate of drug resistant bacteria. We found that 96.7% of the antibiotic-resistant and 92.1% of the AMP-adapted lines display reduced growth compared to the wild type. In agreement with prior studies, the fitness costs associated with AMP resistance was moderate compared to the fitness cost of antibiotic-resistant lines (Fig. 1c). This suggests that fitness cost alone is unlikely to explain why some AMPs are not particularly prone to resistance.

Fourth, AMPs less prone to resistance during adaptive laboratory evolution tend to display increased hydropathicity and contain fewer polar and positively charged amino acids (Fig. 2). The exact cellular mechanisms behind this phenomenon remain to be

elucidated in future research. However, we note that these physicochemical properties influence the capacity of several AMPs to adopt an amphipathic conformation upon binding to the negatively charged bacterial outer membrane[51]. As AMP resistance is often achieved via altering the surface charge of the outer membrane, we speculate that AMPs with relatively few positively charged amino acids might be less prone to resistance by altered membrane surface charge. This hypothesis should be tested in more detail and with a larger number of AMPs in the future.

Fifth, we have conducted a global analysis to explore whether artificially amplified genes provide resistance to individual AMPs or antibiotics. This is relevant, as bacterial gene amplification generates extensive genetic variation that can provide antibiotic resistance[37,38]. For each antibiotic and AMP, we identified the MIC provided by the pooled gene overexpression plasmid library,

and compared it to the MIC of the wild-type strain carrying the empty ASKA plasmid. We have demonstrated that gene amplification can have an important contribution to antibiotic resistance. By contrast, the same library has resulted in no increment in resistance level against 8 of the AMPs tested. These AMPs include R8, TPII, and CP1 (Fig. 6a, Supplementary Table 3). It is important to note that we cannot exclude the possibility that overexpression of genes or combinations thereof enhance bacterial growth rate under sublethal concentrations of AMP stress. This issue, along with the in-depth exploration of the intrinsic AMP resistome will be studied in future chemical genomic analyses[52].

Sixth, we have investigated whether AMP resistance genes are available for genetic exchange across bacterial species present in the soil. The soil bacterial community is a rich source of mobile antibiotic resistance genes[53], and several soil bacterial species display AMP resistance[54,55]. Using standard tools of functional metagenomics, we have failed to find DNA segments that would provide resistance against R8, IND, TPII, and CP1 when transferred into E. coli (Fig. 6b). These results are consistent with a prior work showing that resistance genes in the human gut microbiome originating from phylogenetically distant bacteria have only a limited potential to confer AMP resistance in E. coli, an intrinsically susceptible species[56]. One possible reason for these patterns could be that AMP resistance may an intrinsic bacterial property, potentially influenced by a large number of interacting genes. If so, horizontal gene transfer of short genomic fragments would not provide resistance in the recipient bacterial species.

In sum, our analysis indicates that the frequency by which AMP resistance arises is relatively low and even if it occurs, such genetic changes provide only relatively low levels of resistance. Why is it so? We hypothesize two complementary mechanisms that should be studied in future works. First, evolution of AMP resistance may be hindered by the limited availability of resistance mutations. Multi-targeting drugs in which a single compound inhibits more than one target are generally considered as a promising therapeutic strategy against the rise of resistance[57]. It remains to be elucidated whether AMPs less prone to resistance identified in our study target multiple cellular subsystems. Second, the AMP dose range under which resistance can evolve may be limited. Indeed, AMPs significantly differ from antibiotics in their pharmacodynamics properties. In particular, their bactericidal activity is generally characterized by a sharp increase within a narrow dose range, which is in contrast with the pharmacodynamics of many antibiotics[58]. An important limitation of our work is that we studied evolution and fitness costs of AMP and antibiotic resistance in laboratory conditions only. Future works should elucidate the diversity of AMP resistance mechanisms in more realistic clinical settings.

Finally, our work identifies that TPII and CP1 are promising candidates for future therapeutic considerations, for the following reasons. They are active against a wide range of bacterial pathogens with a relatively high activity/toxicity (ATI) index. We note however, that existing data on the toxicity of the tachyplesin antimicrobial family are controversial[48,59–61]. Promisingly, and in agreement with our results, a previous work showed that CP1 has negligible toxicity or haemolytic activity[62]. Furthermore, evolution of resistance against these AMPs is improbable in E. coli. We propose that these AMPs could serve as a promising base for the development of peptide based drug candidates with limited resistance.

## Methods

**Strains**. Unless otherwise indicated, we used E. coli K-12 BW25113 in all high-throughput experiments. To investigate the impact of mutation rate on evolution of AMP resistance, we initiated laboratory evolution with a mutator strain E. coli

mutD5 that carries mutations in the dnaQ gene (L73W and A164V). This strain exhibits a $10^2$–$10^3$-fold increase in mutation rate[63]. Evolution of resistance against TPII and PXB was also studied in four clinically relevant pathogenic strains, including Escherichia coli (ATCC® 25922™) (abbreviated as E. coli ATCC 25922), Salmonella enterica subsp. enterica serovar Typhimurium LT2 (abbreviated as S. enterica LT2), Klebsiella pneumoniae subsp. pneumoniae (Schroeter) Trevisan (ATCC® 10031™) (abbreviated as K. pneumoniae ATCC 10031) and Acinetobacter baumannii Bouvet and Grimont (ATCC® 17978™) (abbreviated as A. baumannii ATCC 17978). We further tested the spectrum of activity of the most promising compounds on a set of 20 ESKAPE pathogens and other clinically relevant strains: E. coli ATCC 25922, Escherichia coli (ATCC® 35218™), Escherichia coli NCTC 13351, Escherichia coli (ATCC® BAA-2469™), Escherichia coli (ATCC® BAA2340™), Shigella sonnei HNCMB 25021, Shigella flexneri HNCMB 20018, Enterobacter cloacae subsp. cloacae (ATCC® 13047™), S. enterica LT2, Salmonella enteritidis HNCMB 10092, K. pneumoniae ATCC 10031, Klebsiella pneumoniae NCTC 13440, Klebsiella pneumoniae subsp. pneumoniae (ATCC® 700603™), A. baumannii ATCC 17978, Pseudomonas aeruginosa (ATCC® 27853™), Enterococcus faecium (ATCC® 700221™), Streptococcus pyogenes ATCC 19615, Staphylococcus aureus subsp. aureus (ATCC® 25923™), Staphylococcus aureus subsp. aureus (ATCC® 29213™), and Staphylococcus aureus subsp. aureus (ATCC® 43300™).

**Antibacterial agents**. Fourteen cationic AMPs were investigated in this study: PROA, Pleuricidin (PLEU), Bactenecin 5 (BAC5), Human beta-defensin-3 (HBD3), IND, cathelicidin antimicrobial peptide LL-37, IND, rabbit CAP18, PEX, peptide glycine-leucine-amide (PGLA), PR39 (a proline-arginine-rich peptide with 39 residues), R8, Cecropin P1 (CP1), Polymyxin B (PXB) and TPII (Supplementary Table 1). All AMPs were synthesized by ProteoGenix except PXB and PROA which were purchased from Sigma-Aldrich. AMP solutions were prepared in sterile water and stored at −80 °C until usage.

Twelve antibiotics were applied in this study: Tetracycline (TET), Doxycycline (DOX), CHL, Erythromycin (ERY), TRM, Ampicillin (APC), Cefoxitin (FOX), Nalidixic acid (NAL), Ciprofloxacin (CPR), Nitrofurantoin (NIT), Tobramycin (TOB), and Kanamycin (KAN) (Supplementary Table 2). All antibiotics were ordered from Sigma-Aldrich except for ERY (obtained from AMBRESCO) and DOX (purchased from AppliChem). Upon preparation, each antibiotic stock solution was filter-sterilized and kept at −20 °C until usage.

**Medium**. Unless otherwise indicated, Minimal Salt (MS) medium was used in all experiments, including the laboratory evolution experiments. In line with previous, methodologically relevant laboratory evolution studies, the physical conditions (including the medium) were the same during the course of laboratory evolution and all forthcoming assays on the resulting resistant lines (e.g. MIC measurements). We previously optimized this medium for studying AMPs (1 g L$^{-1}$ (NH$_4$)$_2$SO$_4$, 3 g L$^{-1}$ KH$_2$PO$_4$ and 7 g L$^{-1}$ K$_2$HPO$_4$ supplemented with 0.1 mM MgSO$_4$, 0.54 μg mL$^{-1}$ FeCl$_3$, 1 μg mL$^{-1}$ thiamine hydrochloride, 0.2% Casamino acids and 0.2% glucose)[19]. This medium is highly controlled and provides a reproducible environment for experimental evolution experiments. All components were obtained from Sigma-Aldrich.

**Laboratory-evolution experiment**. The evolutionary protocol was designed to avoid population extinction and to ensure that populations with the highest level of resistance were propagated further. Starting with a subinhibitory drug concentration (Supplementary Data 8), resulting in ~50% growth inhibition, 10 parallel populations per drug were allowed to grow in Minimal Salts medium for 72 h and 300 r.p.m. E. coli K-12 BW25113 and E. coli mutD5 populations were incubated at 30 °C, while E. coli ATCC 25922, S. enterica LT2, K. pneumonia ATCC 10031, and A. baumannii ATCC 17978 were incubated at 37 °C. After each incubation period, 20 μl of the 370 μl culture was transferred to four new independent wells containing fresh medium and increasing dosages of the antimicrobial agent (0.5×, 1×, 1.5×, and 2.5× the concentration of the previous step). Prior to each transfer, cell growth was monitored by measuring the optical density at 600 nm (OD$_{600}$ value, Biotek Synergy 2 microplate reader). Only populations with the highest drug concentration (and reaching OD$_{600}$ > 0.2) were selected for further evolution. A chess-board layout was used on the plate to monitor potential cross-contamination events. The evolution experiment was continued for 20 transfers, resulting in a total of 138 AMP (two lines were omitted because of limited growing) and 120 antibiotic-adapted E. coli K-12 BW25113 lines. E. coli ATCC 25922, S. enterica LT2, K. pneumonia ATCC 10031, and A. baumannii ATCC 17978 were adapted to PXB and TPII, resulting in 10 populations per peptide for each species. E. coli mutD5 was adapted to TPII, also in 10 parallel evolving populations. Dose-response curves of E. coli K-12 BW25113 strain against AMPs and antibiotics are provided in Supplementary Fig. 2A and 2B, respectively.

**Determination of MIC**. MICs were determined using a standard serial broth dilution technique[64]. Briefly, 11-step serial dilutions were prepared in 96-well microtiter plates with three replicates per strain and antimicrobial compound. Each experiment was repeated at least twice. $5 \times 10^5$ cells/mL were inoculated into each well, and the plates were incubated at 30 °C in case of E. coli K-12 BW25113, and its antibiotic and AMP-adapted lines, or 37 °C in case of the other bacterial species.

Plates were shaken at 300 r.p.m. during incubation for 24 and 18 h, respectively. Cell growth was monitored by measuring the optical density ($OD_{600}$ value, Biotek Synergy 2 microplate reader was used for this purpose). MIC was defined as complete growth inhibition (i.e., $OD_{600} < 0.05$). Relative MIC was calculated as follows: $MIC_{relative} = MIC_{evolved}/MIC_{control}$. Neither pre-incubation (72 h) nor the exact timing of MIC measurement had any major effect on the MIC of AMPs (Supplementary Table 7).

**High-throughput fitness measurements.** Bacterial cell growth was assayed by continuously monitoring the optical density ($OD_{600}$) of liquid cultures using 384-well microtiter plates containing MS medium, the same medium used during the laboratory evolution. Fitness was measured for a total of 60 antibiotic-adapted and the 38 AMP-adapted E. coli K-12 BW25113 lines (at least four lines per antibiotic or AMP). All measurements were performed twice (six technical replicates per experiment). Starter cultures were inoculated into 96-well plates containing MS medium and incubated for 20 h at 30 °C for E. coli K-12 BW25113 and its evolved lines. After this growth period, the $OD_{600}$ of each culture was read and the starting cell number set to a final of 50,000 cells per well to a final volume of 60 µl. A total of 384-well plates were inoculated combining four different 96-well starter plates together: one plate having the unevolved wild-type control as a reference strain in all wells in order to estimate possible variation in growth within-plate measurement causing biases to the reading, and three plates containing the different set of antibiotic or AMP-adapted lines. Cell growth was monitored by measuring the optical density at 600 nm every 5 min ($OD_{600}$ value, Biotek Synergy 2 microplate reader was used for this purpose). The integral data was calculated by the Biotek Gen5 Data Analysis Software between 1 and 24 h in case of E. coli K-12 BW25113 adapted lines. Because of possible influence of temperature equilibration in the first hour of measurement, this time period was excluded from the analysis.

**Screening of a gene overexpression library.** The ASKA library (a complete set of E. coli K-12 ORF archive) includes every E. coli ORF cloned into an expression vector (pCA24N)[65]. Prior to screening, the ASKA plasmid collection was pooled and transformed into E. coli K-12 BW25113 strain[66]. The plasmid pool was created by pooling an equal aliquot of cells, each carrying different members of the ASKA library, followed by plasmid miniprep. Next, the plasmid pool was electroporated into E. coli K-12 BW25113 competent cells. The electroporation resulted in a number of transformants that covered the full library at least 100 times. For the MICs measurements 500 000 cells were inoculated in each well which represents ~100 times coverage of each library member. To determine the changes in the MIC caused by the overexpression of a specific gene, a 96-well microtiter plate was prepared that contained a 12 step concentration gradient of the corresponding AMP or antibiotic in MS medium supplemented with 10 µg mL$^{-1}$ CHL and 500 µM isopropyl-ß-D-thiogalactopyranoside (IPTG). Prior to the measurements the cell population containing the pooled overexpression library or the control strain containing the empty vector were grown in three replicates in LB medium supplemented with 20 µg/mL CHL. As the culture reached high density (i.e., $OD_{600} > 0.8$), expression of the plasmid was induced by 50 µM IPTG for 1 h. Following induction, $5 \times 10^5$ bacterial cells were inoculated in three replicates into each well of the 96-well plate and incubated for 16–18 h at 37 °C. Following incubation the MIC values were determined by measuring optical densities in a plate reader (Biotek Synergy).

**Functional metagenomic screen.** In order to characterize the horizontally transferable resistome of soil bacteria against four AMPs and four antibiotics, we identified small genomic fragments (1.5–5 kb) that decrease drug susceptibility against the tested AMPs or antibiotics when expressed from a multicopy plasmid with an inducible promoter. For the construction of the metagenomic library, an agricultural soil sample was obtained from Kaposvár, Hungary. Soil community DNA was isolated using Quick-DNA™ Fecal/Soil Microbe Miniprep Kit (Zymo Research) according to the manufacturer's instructions. Then, we followed the steps of the library construction protocol described by Kintses et al.[56]. The average insert size was ~2 kb. The size of the library (3.6 Gb) was estimated by multiplying the average insert size by the number of total colony forming units (CFU).

Functional selections for resistance genes were carried out on solid plates containing one of the four AMPs (R8, TPII, CP1, and IND) or one of the four antibiotics (CPR, CHL, TRM, TOB) serving as controls. Instead of the plating assay that is commonly used in the field[67], we applied a modified gradient plate assay[68], where bacteria are exposed to a concentration gradient of the antimicrobial instead of a single concentration. We followed the steps of the functional metagenomic selection protocol described by Kintses et al[56]. Briefly, $2 \times 10^8$ cells were plated out from the thawed stocks of E. coli K-12 BW25113 bearing the metagenomic plasmid library. In this way, each member of the metagenomic library was represented about 10–100 times on each plate. We found this necessary for a good reproducibility of our experiments. Subsequently, plates were incubated at 30 °C for 24 h. For each functional selection, a control plate was prepared where the same number of E. coli K-12 BW25113 was plated out. These cells contained the pZErO-2 plasmid with a random metagenomic DNA insert that has no effect on AMP or antibiotic resistance. This control plate showed the MIC of the AMP/antibiotic without the effect of a resistance plasmid. The empty plasmid was not applicable as

a control because in the absence of a DNA insert the CcdB toxic protein is expressed from the plasmid. In order to isolate the resistant clones from the library plates, sporadic colonies were identified above the MIC level (defined using the control plate) by visual inspection. These clones were collected by scraping them into 2 ml of LB broth and stored subsequently at −80 °C. Plasmid pools from the scraped resistant clones were PCR amplified for subsequent Illumina sequencing. To this aim, first, the plasmid pools were isolated from each metagenomic selection using InnuPREP Plasmid Mini Kit (Analytic Jena). Then, these plasmid pools served as templates for subsequent PCR amplification of the inserts. These PCR reactions and the subsequent Illumina sample preparation and sequencing were performed as described by Kintses et al.[56].

To assemble the Illumina sequencing reads into longer DNA contigs, we have applied a previously established workflow with modifications[46]. We validated and optimized our workflow on a mock sample containing 5 previously sequenced DNA contigs that originate from one of our metagenomic selections. Reads from the MiSeq $2 \times 250$ bp PE sequencing were pre-processed in Genomics Workbench Tool version 9.0 (CLC Bio) including the following steps: reads were first quality trimmed with an error threshold of 0.05. Then overlapping PE pairs where subsequently merged. Merged and unmerged reads were mapped to pZErO-2 reference plasmid, keeping only those reads that did not match the plasmid sequence. Pre-processed data was then randomly down-sampled to 0.5, 1, 2, 4, 8, 16, 32, 64 and 100% with two replicates generated at each sampling point. MIRA 4.0.2 (http://MIRA-assembler.sourceforge.net/docs/DefinitiveGuideToMIRA.html) was then used to perform de novo sequence assembly separately on each replicate. De novo contigs from MIRA were screened for the presence and proper orientation of the PCR oligonucleotides keeping only complete assembly products. Complete contigs were further filtered based on their average coverage values keeping only the top 20% best-covered ones within each sampling point. Contigs from all sampling points were then pooled. Redundant contigs displaying at least 90% sequence identity to each other over at least 80% of their lengths were clustered using NCBI blastp, keeping only one representative sequence per cluster. The remaining contigs were filtered once more based on their average coverage values until only those Illumina contigs remained in the mock sample which have a corresponding counterpart in the reference Sanger sequences with 100% sequence identity. This criterion was satisfied when contigs with the lowest sequence coverage, forming a distinct peak in the density plot of the coverage distributions, were removed. The optimized workflow was applied for the assembly of the DNA contigs coming from the metagenomic selections. To functionally analyse the ORFs on the assembled contigs from the metagenomic selections, ORFs were predicted and annotated using the Prokka suite (version 1.11[69]) with bacterial prediction settings and an e-value threshold of 10−5 (ref. [56]).

**Whole-genome sequencing.** To identify potential mechanisms conferring resistance to AMPs, we selected those peptides to which at least 2 out of the 10 adapted lines reached a minimum of fivefold increase in resistance (MIC). A total of 38 AMP-adapted lines (at least 4 lines per peptide, see Supplementary Data 1) were chosen for whole-genome sequencing. E. coli K-12 BW25113 genomic DNA was prepared with Sigma GenElute™ Bacterial Genomic DNA Kit and quantified using Qubit dsDNA BR assay in a Qubit 2.0 fluorometer (Invitrogen). Two hundred nanograms of genomic DNA was fragmented in a Covaris M220 focused-ultrasonicator (peak power: 55 W, duty factor: 20%, 200 cycles/burst, duration: 45 s) using Covaris AFA screw cap fiber microTUBEs. Fragment size distribution was analyzed by capillary gel electrophoresis using Agilent High Sensitivity DNA kit in a Bioanalyzer 2100 instrument (Agilent) then indexed sequencing libraries were prepared using TruSeq Nano DNA LT kit (Illumina) following the manufacturer's protocol. This, in short, includes repair of DNA fragments, fragment size selection, ligation of indexed adapters and library enrichment with limited-cycle PCR. Sequencing libraries were validated (library sizes determined) using Agilent High Sensitivity DNA kit in a Bioanalyzer 2100 instrument then quantitated using qPCR based NEBNext Library Quant kit for Illumina (New England Biolabs) with a Piko-Real Real-Time PCR System (Thermo Fisher Scientific) and diluted to 4 nM concentration. Groups of 12 indexed libraries were pooled, denatured with 0.1 N NaOH and after dilution loaded in a MiSeq Reagent kit V2–500 (Illumina) at 8 pM concentration. $2 \times 250$ bp pair-end sequencing was performed with an Illumina MiSeq sequencer, primary sequence analysis was done on BaseSpace cloud computing environment with GenerateFASTQ 2.20.2 workflow. Paired end sequencing data were exported in FASTQ file format. The reads were trimmed using Trim Galore (Babraham Bioinformatics) and cutadapt[70] to remove adapters and bases where the PHRED quality value was <20. Trimmed sequences were removed if they became shorter than 150 bases. FASTQC program (https://www.bioinformatics.babraham.ac.uk/projects/fastqc/) was used to evaluate the qualities of original and trimmed reads. The Breseq program was used with default parameters for all samples. The gdtools was used for annotating the effects of mutations and comparing multiple samples. The genbank formatted reference genome BW25113.gb was used as a reference genome in the analysis.

**pORTMAGE-based insertion of resistance-associated mutations.** pORTMAGE[71] method was used to introduce three putative AMP resistance mutations into the wild-type genetic background (E. coli K-12 BW25113). The mutations include basS L194R, mlaD a342del and waaY M1†, and were identified in one of

the human beta-defensin 3 (HBD3) adapted lines. Briefly, individual mutations were constructed via synthetic ssDNA-mediated recombineering using the pORTMAGE3 vector. The 90 nucleotide long ssDNA oligos were designed using the MODEST tool[72], and had complementary sequences to the replicating lagging strand with a minimized secondary structure ($\geq -12$ kcal mol$^{-1}$). Oligonucleotides were prepared using standard desalting by Integrated DNA Technologies (Coralville, IA, USA). Recombineering was performed in the electrocompetent *E. coli* K-12 BW25113 cells carrying pORTMAGE3 (Addgene plasmid ID: 72678). pORTMAGE3 was induced at 42 °C for 15 min to allow for efficient mutation-incorporation while avoiding off-target mutagenesis[71]. Forty microliters of induced electrocompetent cells were transformed with 1 μl of the respective 100 μM mutation-carrying oligo. Cells were recovered in 5 mL Terrific-Broth (TB) media (24 g yeast extract, 12 g tryptone, 9.4 g K$_2$HPO$_4$, and 2 g KH$_2$PO$_4$ per 1 L of water) after electroporation and incubated at 30 °C for 60 min, after which 5 mL Lysogeny-Broth-Lennox (LB$^L$) media (10 g tryptone, 5 g yeast extract, 5 g sodium chloride per 1 L water) was added. After this, cells were incubated at 30 °C overnight. Appropriate dilutions of the cultures were then plated onto LB$^L$ agar plates to form individual colonies and incubated at 30 °C overnight. Individual colonies were screened using allele-specific PCR to identify ones carrying the desired modifications. An allele-specific primer was designed and tested using wild-type colonies employing a gradient PCR protocol using a Bio-Rad CFX96 Touch thermocycler. The PCR annealing temperature for colony screening was then set at a temperature 1 °C higher than the temperature where the last visible fragment could be detected using the wild-type colony after gradient PCR. Candidate colonies were subsequently verified using Sanger capillary-sequencing. Finally, the pORTMAGE3 plasmid was cured from sequence-verified colonies by growing the cells once overnight at 42 °C in antibiotic-free LB$^L$ media.

For a list of oligos and their sequences used for the mutation construction, see Supplementary Data 9.

**Surface charge measurement**. To investigate changes in bacterial cell surface charge, we performed a fluorescein isothiocyanate-labeled poly-L-lysine (FITC-PLL) (Sigma) binding assay. FITC-PLL is a polycationic molecule that is widely used to study the interaction between cationic AMPs and charged lipid bilayer membranes[73]. In brief, cells were grown overnight in MS medium and then washed twice with 1× phosphate-buffered saline (PBS) buffer. The cells were suspended in the PBS buffer to a final OD$_{600}$ of 0.1. The suspension was incubated with 6.5 μg/mL FITC-PLL for 10 min and centrifuged at 5500 r.p.m. for 5 min. The amount of FITC-PLL in the supernatant remaining after bacterial exposure (or no exposure in case of the control) was determined fluorometrically (excitation at 500 nm and emission at 530 nm). The quantity of FITC-PLL molecules bound to the bacterial surface was calculated from the difference between the amount of FITC-PLL in the control (no exposure) and the amount of unbound FITC-PLL (bacterial exposure). The lower the amount of bound poly-L-lysine, the less negatively charged the cell surface is.

**Hemolysis assay**. Human red blood cells (hemoglobin concentration (Hb) 150–160 g L$^{-1}$) were collected from apparently healthy patients in EDTA tubes. 600 μL of EDTA-blood were washed in TBS buffer (10 mM TRIS, 150 mM NaCl) and centrifuged at $1500 \times g$ for 1 min until the supernatant became colorless. The final pellet was diluted to 5 mL with TBS buffer. Hundred microliters of this cell suspension was pipetted into sterile Eppendorf tubes together with twofold serial dilutions of each compound to a final volume of 200 μl. Final concentrations ranged between 2500 μg mL$^{-1}$–9.75 μg mL$^{-1}$. Following incubation for one hour at 37 °C, samples were centrifuged at 1500 g for 1 min to precipitate the red blood cells. All supernatants were transferred to sterile 96-well plates for the measurement of their direct optical density (OD) at 565 nm wavelength (Multiskan FC microplate reader, Thermo Scientific). Melittin (Bachem) at concentration of 50 μg. mL$^{-1}$ and TBS were used as positive (100% hemolysis) and negative (no hemolysis) controls, respectively. Haemolytic effect of each peptide at each concentration was calculated as follows: Hemolysis effect = (Compound OD$_{565nm}$ − TBS OD$_{565nm}$) × 100/(Melitin OD$_{565nm}$ − TBS OD$_{565nm}$).

**Reporting summary**. Further information on research design is available in the Nature Research Reporting Summary linked to this article.

## Data availability

The authors declare that the main data supporting the findings of this study are available within the article and its Supplementary Information files. Dose-response curves of AMP-adapted lines are available from the corresponding author upon request. The source data underlying Figs. 1a–c, 2a–c, 5, 6a–b and Supplementary Figs. 1 and 2a–b are provided as a Source Data file. The whole-genome sequencing data can be accessed from the NCBI GeneBank with access number: PRJNA555839 and the functional metagenomics sequencing data can be accessed from MN256796-MN256836.

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

## Acknowledgements

The study was supported by the following research grants: The European Research Council H2020-ERC-2014-CoG 648364 – Resistance Evolution (CP); National Research, Development and Innovation Office, Hungary, NKFIH grant K120220 (BK), NKFIH grant FK124254 (OM) and NKFI PD 116222 (AM), OTKA PD 109572 (BC) and 'Lendület' Programme of the Hungarian Academy of Sciences (C.P. and Ba.P.); The Wellcome Trust (098016/Z/11/Z for BP and 084314/2/07/2 for C.P.); 'Célzott Lendület' Programme of the Hungarian Academy of Sciences LP-2017–10/2017 (CP); 'Élvonal' KKP 126506 (C.P.), GINOP-2.3.2–15–2016–00014 (EVOMER, for C.P. and Ba.P.), GINOP-2.3.2.–15–2016–00020 (MolMedEx TUMORDNS) and GINOP-2.3.3–15–2016–00001. B.K. holds a Bolyai Janos Scholarship and supported by the UNKP-18–4 New National Excellence Program of the Ministry of Human Capacities. OM holds a Bolyai János Research Fellowship from the Hungarian Academy of Sciences. R.W., G.M., and Be.P. receive support from projects PD121085, FK123899 and KH125616 provided by the HNRDIF, financed under the PD16, FK17, and KH17 funding schemes, respectively. LB was supported by grant NKFI-112294. GM hold Bolyai Janos Scholarship. The authors thank Dora Bokor for proofreading the manuscript.

## Authors contributions

C.P. conceived and supervised the project; R.S., L.D., V.L., A.M., and Z.B. performed the laboratory evolution experiment; R.S., L.D., A.M., M.S., and F.V. performed the minimum inhibitory concentration (MIC) measurements; O.M., B.K., M.S., and A.G. performed the functional metagenomic screen; G.M., Be.P., and R.W. provided the metagenomic DNA isolated from the soil sample; A.F. and L.B. performed the whole-genome sequencing; L.D., F.V. performed the fitness measurements; L.D., M.S., and F.V performed the gene over-expression measurements; P.K.J. performed the surface charge experiments; B.C. prepared the single mutant strains; I.F. and D.K. performed the hemolysis assay; R.S., L.D., G.G., A.G., C.P., and B.aP. analyzed and interpreted the data. C.P., B.aP., R.S., L.D., and A.M. wrote the manuscript. All authors have read and approved the manuscript, and the corresponding author had the final responsibility to submit the study for publication.

## Competing interests

The authors declare no competing interests.
