## [Peer Review File · Nature Communications]

Reviewers' Comments:

Reviewer #1:

Remarks to the Author:

Antimicrobial peptides (AMPs) have for a long time been put forward as potential alternative antibiotics. However, development has been hampered by factors such as high toxicity, susceptibility to proteases, and high cost of peptide production. Moreover, there has been a general problem of separating toxicity and antimicrobial activity by structural manipulation of the peptides where increased activity often goes hand in hand with increased toxicity. These factors have contributed to the lack of success in transforming antimicrobial peptides into clinically relevant antimicrobials. Here the authors systematically investigate the potential of 14 structurally diverse peptides to select resistance in K12 BW25113. This ability was then compared to a set of small molecule antibiotics of clinical relevance. This work is novel in at least two ways. Firstly, it systematically investigates the resistance potential of a large number of AMPs and it does that with a combination of novel methodological approaches.

The paper shows that towards some AMPs resistance is less likely compared to conventional antibiotics but also other AMPs. In particular, two peptides seem to be remarkably difficult for bacteria to evolve resistance against. For peptides where resistance evolved the molecular mechanisms underlying resistance is investigated through whole genome sequencing of resistant clones, fitness cost and mechanisms are investigated and the potential of cross-resistance investigated. Moreover, the surface charge of the resistant mutants were investigated and if gene amplification could have an important influence on resistance evolution. For the two peptides with the least resistance potential the in vitro toxicity was also studied.

Lastly, an attempt to investigate the potential of the soil microbiome as a reservoir for resistance genes that could be horizontally transferred was made through functional genomics with the conclusion that this may be rare.

The manuscript is exceptionally well written, where methodology, results and conclusions are clearly described.

This is, to my knowledge, the first systematic investigation of resistance potential of a large number of AMPs. Moreover, the paper presents some methodological innovations that develop the field of experimental evolution and how resistance mechanisms are studied.

This work contributes new information that may aid the development of AMPs into clinically relevant antibiotic alternatives but also provides a novel systematic approach regarding the in vitro study of resistance evolution.

Major comments,

The abstract could be more descriptive of the study. It would be informative to include information such as number of AMPs and antibiotics that have been tested. The investigation of resistance development in other pathogens and the comparison of AMPs to other antibiotics could be relevant to describe.

The specific result that peptides such as Tachyplesins and cecropins have low resistance evolution potential are not novel in themselves. These classes of AMPs have been under intense scrutiny for many years, a more comprehensive discussion of what is known about these molecules regarding resistance evolution, and pharmacological properties should be included. This is particularly important regarding toxicity and what is known about resistance towards this class of molecules. Here the activity/toxicity index is of particular interest and the authors should make an effort to explain why this is the most interesting measurement when this somewhat goes against what has been published previously. Moreover, Tachyplesins and cecropins are sensitive to protease degradation. It is therefore somewhat surprising that the functional genomics investigation finds any resistance determinants. The functional genomics part is somewhat redundant. I do not think this approach is very valuable to give a robust indication of horizontal transfer potential. It is rather well established that resistance towards

many peptide are multifactorial and complex. Expressing gene libraries in E.coli give a too narrow a picture because most DNA from gram-positive organisms will not be expressed for example. It does therefore only superfluously the conclusions of the work.

All this taken in consideration I think that this manuscript I should be published and will be an important contribution in the field of development of novel antibiotics and experimental evolution.

Specific comments:

- Why is the evolution experiment an "automated laboratory experiment"? Based on the description in the "Method" section it is a serial transfer of the incubated cultures into increasing concentrations of AMPs and antibiotics. What does "automated" refer to?
- L2: What does "integrated evolutionary analysis" cover? This concept is not explained or defined in the manuscript which could be misleading. I would suggest clarification or rephrasing
- L110: What happened to 2 of the AMP adapted lines? 140 AMP-adapted lines were made but only 138 are used in Fig 1 and further on. Did 2 of them extinct?
- L135: it would be interesting to see if a mutator strain is able to become resistant to R8.
- L136: change first word "strikingly" in either this line or line 131.
- L216: dN/dS ratio could support the statement about positive selection.
- L229: It is not clear why HBD3 is included.
- L263-264: It is unclear how and what MIC value has been used for the selection. It is stated in the "Method" section but it would be good to make a comment on it for clarity of the selection procedure. How many clones have been investigated, i.e. is the number of investigated clones enough to assume that all genes have been investigated?
- Figure 5: it is interesting that the same mutations give such a large variation in the surface charge. In line 214-216 the number of SNPs and indels is stated. Compared to that it is surprising that there are so many secondary mutations, which can explain the large variation in the surface charge.
- Figures and tables
- Figure 5: it is interesting that the same mutations give such a large variation in the surface charge. In line 214-216 the number of SNPs and indels is stated. Compared to that it is surprising that there are so many secondary mutations, which can explain the large variation in the surface charge.
- L476: insert space E.coli.
- Materials and methods
- In general: changing ug.mL⁻¹ to µg/mL⁻¹, would be appreciated.
- L802: erase line.
- L957: cca, correct.
- L963: ug.mL⁻¹, correct superscript.
- L967: same as above.
- L968: erase space before %
- L791: µg/mL instead of µg.mL⁻¹
- L796-798: why is IPTG not added for the MIC determination?

Reviewer #2:

Remarks to the Author:

The authors compare the ability of various bacterial strains to evolve resistance to AMPs antibiotics, and also look at over-expression of genes, and horizontal gene transfer. I think the experimental methodology is good, and I especially like the tests of overexpressing plasmids and inserting environmental DNA. The results are novel, though perhaps the clinical importance is vaguely justified.

However, I think in many cases the statistical analyses the authors present are insufficient. In my

opinion, considerable changes need to be made to the analyses. Primarily, often multiple two-sample tests are used where the use of factorial analysis designs (e.g. <https://www.nature.com/articles/nmeth.3180>) would be more appropriate. The authors use non-parametric statistics on ranks (Mann-Whitney U and Spearman's rho) without explanation, but state conclusions on the data rather than the ranked data, and sometimes pool treatments without justification (e.g. AB vs AMPs in figure 6a).

On methods, some information needed to reproduce experiments (i.e. AMP and AB concentrations used) are not given. It would be nice to see dose-response curves for the AMPs especially---without which I think it would be hard to directly compare evolution in the face of changing relative concentrations. Perhaps a 2x increase for TPII is worse than a 2x increase in PXB.

Specific comments:

Abstract

L38 of the abstract says 'no cross-resistance', but surely 13.4% is bigger than none?

L39 'gene amplification, an important genetic source of antibiotic resistance'---while true for some antibiotics, this isn't necessarily supported by the data shown in the paper, in which large changes in MIC seem to be shown primarily for 3 of the drugs, trimetoprim, ampicillin and cefoxitin

The abstract and discussion mention a lack of 'clinically relevant' resistance to AMPs in experimental evolution. The paper could use a clearer definition of what clinically relevant resistance for AMPs would look like.

Methods

I think some information that would be needed for reproducing the experiments is missing or in a place that was difficult to find. I cannot find what absolute concentrations of either AMPs or ABs were used during selection. Relative concentration steps are mentioned on L737.

L730 mentions 'a subinhibitory concentration, resulting in ~50% growth inhibition', but ideally information on exact concentrations would be included here. Pointing to a supplementary table in the methods ought to suffice---perhaps in Table S1 (though I also found no information in the supplement).

Results

L198: Why is this 'only' 13.4% cross resistance? The authors could make more clear what their benchmark for cross-resistance is. L38 of the abstract says 'no cross-resistance', but surely 13.4% is bigger than none (as mentioned above)?

Figure 1: I don't think it's appropriate to do a Mann-Whitney U test on pooled ABs vs AMPs (panel a), as there is heterogeneity in both (as shown in panel b)

Figure 2: Spearman's rho (a non-parametric rank correlation, i.e. the Pearson correlation of the ranks) is given in the caption, but then a log-linear trend is plotted through the data. If a linear trend is shown, it would be more appropriate to show a Pearson correlation on the data, though I'm not particularly convinced a linear relationship is appropriate, especially for hydrophobicity, but I'm not sure it adds anything for any of these plots.

Figure 3: it's unclear what the underscores followed by a number mean here---do these refer to lineages evolved in that AMP?

Figure 4: similarly unclear what _numbers mean. Conventionally, gene names should be in italics. "to a given AMPs" should be "AMP" singular?

Figure 5: Should the use of multiple tests not merit a correction for multiple comparisons, e.g. Bonferroni, which at the 'conventional' $\alpha = 0.05$ would result in a corrected α of $0.05/20 = 0.0025$. I'm also confused by the side-table and the bold part of the caption. I thought the caption suggests that lines carry a single mutation, but the table seems to show some strains with mutations in more than one gene?

Figure 6: There are 3 'outliers' for antibiotics: trimethoprim, ampicillin and cefoxitin. I do not think it is appropriate to bin the antibiotics into a category and conduct a two-way test on AMPs vs Abs---clearly there is heterogeneity in resistance to ABs conferred by overexpression. I think a nested ANOVA design here would seem more appropriate (see e.g. <https://www.nature.com/articles/nmeth.3137>).

Results:

L415: 'relatively immune to resistance' is vague, as it does not specify relative to what---to other AMPs? ABs? I wonder if a different word could be used instead of 'immune'---can something be 'relatively immune'?

Minor:

L356: remove the extra %

L358: associated to associated with?

L363: less fewer? (If referring to the #/proportion of amino acids)

L795: The definition of IPTG is given here, but comes after IPTG is first used in the text

The usage of species names is non-standard in parts (i.e. full genus and species for first usage, and abbreviated genus afterward) but hopefully that can be addressed by typesetting. Similarly, in places both K12 and K-12 are used (I believe it's *Escherichia coli* str. K-12 substr. BW25113 'officially', for the others I'm not sure).

Throughout the manuscript could use checking that terms are used consistently.

Reviewer #3:

Remarks to the Author:

This manuscript from Spohn et al involves a study of the evolution of resistance to a diverse set of AMPs and antibiotics in *Escherichia coli*. Based on several different types of experimental data they suggest that AMPs are less prone to resistance development than classical small molecule antibiotics for four reasons: (i) in laboratory evolution experiments AMPs are less prone to resistance evolution than antibiotics, (ii) limited cross-resistance is seen between AMPs, (iii) increased copy number of native genes cannot confer AMP resistance and (iv) soil meta-genomic fragments cannot confer AMP resistance.

Overall I think this is a nice study with some important findings that are suitable for Nature Communications. However, there are a few things the authors need to address before it would be acceptable.

1) The choice of soil metagenomics as a source for AMP-resistance genes is unclear to me. Are AMPs present in soil? If not, what was the reason to choose soil over other microbiomes that are more likely to encounter AMPs? Some more discussion on this would be helpful to follow the authors motivation

for this particular experimental setup.

2) A table should be provided that give the starting MIC values for the all species/drug combinations that have been tested. This data should already be available to the authors since the MIC's must have been determined to define the starting concentrations for the evolution experiment.

3) Why were some strains grown at 30°C and others at 37°C?

4) One major claim of the study is that resistance is not easily selected in vitro during an evolution experiment with increasing AMP-concentration. The authors should provide more information regarding the specific conditions of the cycling, for example how many cells were transferred during each cycling, how many generations of growth were achieved per cycle and what is the final cell density in the described MS-media. Using this information, it would also be very interesting to model what the minimal rate of emergence of potential rare resistance mutation should be to be selected under these experimental conditions.

5) Cells were grown for 72h each cycle. Are active AMP concentrations constant during this time? It is known that some AMP have limited stability under laboratory conditions and concentrations might be reduced during incubation, for example, by absorption to plastic). Thus, could it be possible that the active concentration decreases which subsequently prevents enrichment of mutants? This might be especially important when the pharmacodynamics response curve is very steep and the selective window is narrow (see also point 6 below). An easy way to address this would be by two separate microdilution MIC determinations, one with freshly prepared media, and one with media (+AMP) that was pre-incubated for 72h. Ideally, the MIC should be identical for both assays.

6) I am glad to see that the authors discuss the paper (ref 53) regarding the steeper pharmacodynamics response and narrower selective window of AMPs compared to antibiotics. I suspect that this is a main factor that reduces the risk of AMP resistance development in clinical settings. Perhaps the author could in the discussion, in addition to the 4 reasons they list in the Abstract for why AMP resistance evolution is slower compared to antibiotics, explicitly include the PD aspects as a 5th reason. Also, it would have been wonderful if the authors had included such PD data for the AMPs and antibiotics studied (however it is not something necessarily needed for this paper).

7) The authors show that the fitness costs associated with AMP resistance mutations is lower than that associated with antibiotic resistance mutations. However, I am not sure if one can draw the conclusion from this that resistance mutations for the specific AMPs are or are not observed because of being costly/detrimental. The analysis, as I understand it, does not include the mutants observed for CP1, PGLA, IND and PLEU, which are among the AMPs against which development of resistance is limited. I understand that the authors are trying to extrapolate by making fitness cost a general phenotype, but I am not sure one can do that. Also, the authors should also mention that these costs are on isolated mutants and not on reconstructed mutants, and might thus have other mutations present as well.

8) The authors write "...Reassuringly, 94% of all point mutations were non-synonymous, indicating that their accumulation was driven by positive selection." I am not sure if accumulation of non-synonymous mutations over only 120 generations can be used to make this argument, especially because one generally expects to observe many more non-synonymous mutations in such experiments.

9) The type of reconstructed mutations in BasS, WaaY and mlaD should be specified, especially if there is any knowledge of whether these are loss of function mutations are not.

10) The authors should discuss more about the MICs of CPI and TPII for *Enterobacter cloacae* (Supplementary table 11), since they are much higher than for the rest of the Gram-negatives tested.

Response to reviewers

First of all, we wish to thank the reviewers for their insightful comments. We attempted to address all remaining concerns. Please find our detailed responses below.

Reviewers' comments:

Reviewer #1 (Remarks to the Author):

Antimicrobial peptides (AMPs) have for a long time been put forward as potential alternative antibiotics. However, development has been hampered by factors such as high toxicity, susceptibility to proteases, and high cost of peptide production. Moreover, there has been a general problem of separating toxicity and antimicrobial activity by structural manipulation of the peptides where increased activity often goes hand in hand with increased toxicity. These factors have contributed to the lack of success in transforming antimicrobial peptides into clinically relevant antimicrobials. Here the authors systematically investigate the potential of 14 structurally diverse peptides to select resistance in K12 BW25113. This ability was then compared to a set of small molecule antibiotics of clinical relevance. This work is novel in at least two ways. Firstly, it systematically investigates the resistance potential of a large number of AMPs and it does that with a combination of novel methodological approaches.

The paper shows that towards some AMPs resistance is less likely compared to conventional antibiotics but also other AMPs. In particular, two peptides seem to be remarkably difficult for bacteria to evolve resistance against. For peptides where resistance evolved the molecular mechanisms underlying resistance is investigated through whole genome sequencing of resistant clones, fitness cost of mechanisms are investigated and the potential of cross-resistance investigated. Moreover, the surface charge of the resistant mutants were investigated and if gene amplification could have an important influence on resistance evolution. For the two peptides with the least resistance potential the in vitro toxicity was also studied.

Lastly, an attempt to investigate the potential of the soil microbiome as a reservoir for resistance genes that could be horizontally transferred was made through functional genomics with the conclusion that this may be rare.

The manuscript is exceptionally well written, where methodology, results and conclusions are clearly described. This is, to my knowledge, the first systematic investigation of resistance potential of a large

number of AMPs. Moreover, the paper presents some methodological innovations that develops the field of experimental evolution and how resistance mechanism are studied. This work contribute new information that may aid the development of AMPs into clinically relevant antibiotic alternatives but also provides a novel systematic approach regarding the in vitro study of resistance evolution.

Thank you for your interest in our work.

Major comments,

The abstract could be more descriptive of the study. It would be informative to include information such as number of AMPs and antibiotics that have been tested. The investigation of resistance development in other pathogens and the comparison of AMPs to other antibiotics could be relevant to describe.

We modify the abstract as follows: Here we systematically study the evolution of resistance to 14 chemically diverse AMPs and 12 antibiotics in *Escherichia coli*.

The specific result that peptides such as Tachyplesins and cecropins have low resistance evolution potential are not novel in themselves. These classes of AMPs have been under intense scrutiny for many years, a more comprehensive discussion of what is known about these molecules regarding resistance evolution, and pharmacological properties should be included. This is particularly important regarding toxicity and what is known about resistance towards this class of molecules. Here the activity/toxicity index is of particular interest and the authors should make an effort to explain why this is the most interesting measurement when this somewhat goes against what have been published previously. Moreover, Tachyplesins and cecropins are sensitive to protease degradation. It is therefore somewhat surprising that the functional genomics investigation find any resistance determinates.

Existing data on the toxicity of the tachyplesin antimicrobial family are controversial (Liu et al 2018 PMID: 29765362, Ramamoorthy et al., 2006 but Cirioni et al., 2007, Edwards et al., 2017). This was an important motivation for studying activity/toxicity index explicitly, as described by Edwards et al. (2017 ACS Infect Dis.) Please note also some promising results and antimicrobial activities with TPIII in an in vivo sepsis mouse model (Cirioni et al., 2007).

As regards CP1, the cytotoxicity and the haemolytic activity of this peptide has been investigated (PMID: 29282543). In agreement with our results, this study demonstrated that CP1 had negligible toxicity or haemolytic activity.

Finally, our manuscript provides a detailed analysis on TPII and CP1 being a potential resistance-free parental peptide. We do not claim that they could be used directly for clinical applications, but could be starting points of developing antimicrobial peptidomimetics.

The functional genomics part is somewhat redundant. I do not think this approach is very valuable to give a robust indication of horizontal transfer potential. It is rather well established that resistance towards many peptide are multifactorial and complex. Expressing gene libraries in *E.coli* give a too narrow a picture because most DNA from gram-positive organisms will not be expressed for example. It does therefore only superfluously the conclusions of the work.

Functional metagenomics is an established, systematic approach specifically developed to explore the reservoir of mobile resistance genes. However, we agree with the reviewer concerns, and added a short text in the discussion as follows: “These results are consistent with a prior work showing that resistance genes in the human gut microbiome originating from phylogenetically distant bacteria have only a limited potential to confer AMP resistance in *E. coli*, an intrinsically susceptible species⁵¹. One possible reason for these patterns could be that AMP resistance may an intrinsic bacterial property, potentially influenced by a large number of interacting genes. If so, horizontal gene transfer of short genomic fragments would not provide resistance in the recipient bacterial species.”

All this taken in consideration I think that this manuscript I should be published and will be an important contribution in the field of development of novel antibiotics and experimental evolution.

Specific comments:

- Why is the evolution experiment an “automated laboratory experiment”? Based on the description in the “Method” section it is a serial transfer of the incubated cultures into increasing concentrations of AMPs and antibiotics. What does “automated” refer to?

We agree that it was not clear-cut. We used specific scripts, pipelines and laboratory robots to minimize human intervention during laboratory evolution and high-throughput MIC assays. To avoid any further confusion, we delete the term “automated” throughout the manuscript.

- L2: What does “integrated evolutionary analysis” cover? This concept is not explained or defined in the manuscript which could be misleading. I would suggest clarification or rephrasing

We believe that most prior pioneering studies on AMP resistance were of limited scope: They 1) focused on a handful, functionally closely related AMPs 2) and/or one bacterial species only. They 3) studied mutational resistance only, and largely ignored AMP resistance by horizontal gene transfer. Finally, and most importantly, 4) no prior studies attempted to compare the rate of resistance evolution against AMPs and clinically employed antibiotics systematically. We attempted to overcome these shortcomings by integrating the results of several complementary assays.

To explain these issues better, we write in the introduction: “Here we systematically characterize bacterial potential to acquire resistance against a set of chemically diverse AMPs (Supplementary Table 1). For this purpose, we have combined laboratory evolution, systematic gene overexpression studies and functional metagenomics. We integrated the results of these complementary tests to get a global overview of AMP resistance evolution.”

- L110: What happened to 2 of the AMP adapted lines? 140 AMP-adapted lines were made but only 138 are used in Fig 1 and further on. Did 2 of them extinct?

In these two specific cases, we have observed limited or no growth of bacterial populations from a certain time points during the course of laboratory evolution. For this reason, they were omitted from the analysis.

- L135: it would be interesting to see if a mutator strain is able to become resistant to R8.

We agree. As a matter of fact, we plan to devote a full work on R8 resistance evolution in a future work.

- L136: change first word “strikingly” in either this line or line 131.

Done

- L216: dN/dS ratio could support the statement about positive selection.

To statistically test whether the ratio of non-synonymous to synonymous SNPs was higher than expected based on a neutral model of evolution, we employed an established method (Barrick et al. Nature 2009), that is especially well-suited for experimental evolution studies with limited

number of observed mutations (Szamecz et al. Plos Biology 2014). Briefly, we took all different point mutations observed in protein coding regions and calculated the probability that 94% or more substitutions would result in a non-synonymous substitution if it occurred in a random coding position. The excess of non-synonymous substitution observed in the evolved genomes was significant ($p = 0.000004$).

- L229: It is not clear why HBD3 is included.

HBD3 is a clinically relevant antimicrobial peptide with human origin.

- L263-264: It is unclear how and what MIC value has been used for the selection. It is stated in the “Method” section but it would be good to make a comment on it for clarity of the selection procedure. How many clones have been investigated, i.e. is the number of investigated clones enough to assume that all genes have been investigated?

Prior to screening, the ASKA plasmid collection was pooled and transformed into *E. coli* K-12 BW25113 strain as described previously (Notebaart, R. A. et al., PNAS 2014). Briefly, the plasmid pool was created by pooling an equal aliquot of cells, each carrying different members of the ASKA library, followed by plasmid miniprep. Next, the plasmid pool was electroporated into *E. coli* competent cells. The electroporation resulted in a number of transformants that covered the full library at least 100 times. For the MICs measurements 500 000 cells were inoculated in each well which represents ~100 times coverage of each library member. “

For the used MIC values please see Supplementary table 9.

- Figures and table
- Figure 5: it is interesting that the same mutations give such a large variation in the surface charge. In line 214-216 the number of SNPs and indels is stated. Compared to that it is surprising that there are so many secondary mutations, which can explain the large variation in the surface charge.

Figure 5 depicts repeatedly mutated genes potentially influencing surface charge. Accordingly, the variation may reflect the accumulation of *different* mutations in the same genes (such as BasS, WaaY), or as noted by the reviewer the action of other mutated genes. We briefly discuss the latter possibility:

“However, it is probable that many other mutated genes contribute to the observed changes in negative surface charge of the outer membrane. “

- L476: insert space E.coli.
- Materials and methods
- In general: changing ug.mL-1 to ug/mL-1, would be appreciated.
- L802: erase line.
- L957: cca, correct.
- L963: ug.mL-1, correct superscript.
- L967: same as above.
- L968: erase space before %
- L791: µg/mL instead of µg.mL-1

All these typos were corrected, thank you for raising our attention to them.

- L796-798: why is IPTG not added for the MIC determination?

IPTG was added as we mention it in line 791.

Reviewer #2 (Remarks to the Author):

The authors compare the ability of various bacterial strains to evolve resistance to AMPs antibiotics, and also look at over-expression of genes, and horizontal gene transfer. I think the experimental methodology is good, and I especially like the tests of overexpressing plasmids and inserting environmental DNA. The results are novel, though perhaps the clinical importance is vaguely justified.

However, I think in many cases the statistical analyses the authors present are insufficient. In my opinion, considerable changes need to be made to the analyses. Primarily, often multiple two-sample tests are used where the use of factorial analysis designs (e.g. <https://www.nature.com/articles/nmeth.3180>) would be more appropriate. The authors use non-parametric statistics on ranks (Mann-Whitney U and Spearman's rho) without explanation, but state conclusions on the data rather than the ranked data, and sometimes pool treatments without justification (e.g. AB vs AMPs in figure 6a).

Thank you for the observation. Although we couldn't apply nested ANOVA due to large differences in the variance between drugs, we did perform a permutation test to address this issue. We altered the tests used in the comparison of the MIC fold changes for both figure 1 and 6.

On methods, some information needed to reproduce experiments (i.e. AMP and AB concentrations used) are not given. It would be nice to see dose-response curves for the AMPs especially---without which I think it would be hard to directly compare evolution in the face of changing relative concentrations. Perhaps a 2x increase for TPII is worse than a 2x increase in PXB.

We show the requested dose response curves, see Supplementary Figure 2A and B.

Specific comments:

Abstract

L38 of the abstract says 'no cross-resistance', but surely 13.4% is bigger than none?

The cited sentence was as follows: Second, drug-resistant bacteria have displayed no cross resistance to these AMPs. This statement is indeed ambiguous, as it was strictly true for only three AMPs, R8, TPII and CP1 (Figure 3, Supplementary Table 6). We modify it as follows:

“Resistance level provided by point mutations and gene amplification is very low and antibiotic resistant bacteria display no cross resistance to these AMPs.”

L39 'gene amplification, an important genetic source of antibiotic resistance'---while true for some antibiotics, this isn't necessarily supported by the data shown in the paper, in which large changes in MIC seem to be shown primarily for 3 of the drugs, trimetoprim, ampicillin and ceftiofur

We modify it: as a potential source of antibiotic resistance.

The abstract and discussion mention a lack of 'clinically relevant' resistance to AMPs in experimental evolution. The paper could use a clearer definition of what clinically relevant resistance for AMPs would look like.

We agree that it is difficult to define a threshold for clinically significant resistance. Therefore, we modify the abstract as follows: “ Resistance level provided by point mutations and gene

amplification is very low, and antibiotic resistant bacteria display no cross resistance.” We also modified the discussion to avoid the term “clinically relevant resistance”.

Methods

I think some information that would be needed for reproducing the experiments is missing or in a place that was difficult to find. I cannot find what absolute concentrations of either AMPs or ABs were used during selection. Relative concentration steps are mentioned on L737. L730 mentions 'a subinhibitory concentration, resulting in ~50% growth inhibition', but ideally information on exact concentrations would be included here. Pointing to a supplementary table in the methods ought to suffice—perhaps in Table S1 (though I also found no information in the supplement).

We agree that this information is necessary, and it can now be found in Supplementary table 14.

Results

L198: Why is this 'only' 13.4% cross resistance? The authors could make more clear what their benchmark for cross-resistance is. L38 of the abstract says 'no cross-resistance', but surely 13.4% is bigger than none (as mentioned above)?

We agree. This statement only holds for TPII, R8 and CP1, and we modified the abstract accordingly. We deleted the word “only” and modified the paragraph as follows:

“Three main findings have emerged from the analysis. First, 13.4% of all possible combinations of the AMPs and the evolved lines have displayed cross-resistance (Figure 3, Supplementary Table 6). Second, only 5 out of the 33 non-LL37 evolved lines have shown cross-resistance to the human peptide LL37. Moreover, 4 out of these 5 evolved lines had been exposed to CAP18 during the course of laboratory evolution. As CAP18 belongs to the cathelicidin family³², chemical relatedness may drive cross-resistance to AMPs. Third, and most significantly, none of the 38 AMP-resistant lines displayed cross-resistance to R8, TPII and CP1 (Figure 3, Supplementary Table 6). These results further confirm that resistance mutations against these specific AMPs are particularly rare.”

Figure 1: I don't think it's appropriate to do a Mann-Whitney U test on pooled ABs vs AMPs (panel a), as there is heterogeneity in both (as shown in panel b).

In Figure 1, instead of using pooled data, we calculated of the mean log₁₀ transformed MIC values per evolved strains or drugs (figure 1). Next, we used a permutation test to see whether the antibiotic groups show significantly higher MIC values than the AMP groups. In the test, we randomly assigned sets of parallel evolved lines to drug treatments and calculated the difference in mean MIC values between AMP and antibiotic-evolved sets. This procedure was repeated 10000 times. In both cases, we found, that the real difference between the mean values of the two groups was significantly higher than the differences of the means calculated after permutation ($p = 0.0007$). We modified the figure legend as follow:

Figure 1-Minimal inhibitory concentration (MIC) and relative fitness of adapted lines after the laboratory evolution. a) Relative resistance level in laboratory evolved *E. coli* K-12 lines exposed to one of each 14 AMPs or 12 antibiotics, respectively (at least 9 parallel evolved lines per drug). Altogether, lines have been exposed to AMPs (N=138) developed significantly lower resistance, than lines have been exposed to antibiotics (N=120) ($P < 0.0001$, permutation test). The resistance levels reached were more heterogeneous across AMP treatments (N=14) than across antibiotic treatments (N=12) (F-test, $P = 0.03478$). The mutD5 mutator strain exposed to TPII is marked by an asterisk (*). b) Resistance level after laboratory evolution in clinical isolates under TPII or PXB stresses, respectively. Evolved lines exposed to TPII reached significantly lower resistance level than the lines exposed to PXB (*) indicate the significant difference at least $P\text{-value} = 1.65 \times 10^{-4}$, two-sided Mann-Whitney test, $N = 10$ each group). c) Relative fitness of 60 antibiotic-resistant and 38 AMP-resistant lines displaying at least 2-fold increments in resistance level to the drug indicated. Fitness was measured as the area under the growth curve in an antibacterial agent-free medium and was normalized to that of the wild-type (grey colour). Throughout Figure 1, boxplots show the median, first and third quartiles, with whiskers showing the 5th and 95th percentile. For AMP and antibiotic abbreviations, see Supplementary Tables 1-2.**

Figure 2: Spearman's rho (a non-parametric rank correlation, i.e. the Pearson correlation of the ranks) is given in the caption, but then a log-linear trend is plotted through the data. If a linear trend is shown, it would be more appropriate to show a Pearson correlation on the data, though I'm not particularly convinced a linear relationship is appropriate, especially for hydrophobicity, but I'm not sure it adds anything for any of these plots.

We now use the LOESS (locally estimated scatterplot smoothing) method for curve fitting, and modified the figure legend as follow:

Figure 2-Correlation between AMPs' physicochemical features and AMP resistance level. Each datapoint shows the average MIC-fold change in laboratory evolved *E.coli* K-12 lines exposed to one of each 14 AMPs. a) Fraction of polar amino acids and relative resistance level (Spearman's $\rho=0.58$; $p=0.03$, $N=14$), b) Fraction of positively charged and relative resistance level (Spearman's $\rho=0.62$; $p=0.02$, $N=14$), and c) AMP Hydrophobicity and relative resistance level (Spearman's $\rho=-0.73$ $p=0.002$, $N=14$). For AMP properties, see Supplementary table 5. Blue lines indicate the curve fitted using LOESS smoothing method in R.

Figure 3: it's unclear what the underscores followed by a number mean here---do these refer to lineages evolved in that AMP?

Yes, exactly. The names of AMP-resistance lines are composed from the abbreviation of the AMP to which the *E. coli* has been exposed in the evolutionary experiment, and after the underscore the serial number of the AMP-resistance line from the 10 independent populations. We modified the figure legends as follows:

Figure 3 – Cross-resistance of AMP-resistant lines (rows) towards a set of 7 AMPs (columns). Relative minimum inhibitory concentration (MIC) was calculated as the ratio of the MIC of the resistant line and the sensitive wild-type strain. Hierarchical clustering was performed separately on rows and columns, using complete linkage method with Euclidean distance measure on the raw MIC data. Throughout the figure, blue coloring refers to collateral-sensitivity (MIC two-fold lower than the wild-type), orange coloring refers to cross-resistance (MIC two-fold higher than the wild-type), white coloring refers to no or minimal change in susceptibility (MIC in between). Grey coloring refers to not applicable. AMP-resistant lines are named after the AMP it has been exposed to during the evolutionary experiment, and the serial number of the corresponding population. For AMP abbreviations, see Supplementary Table 1.

Figure 4: similarly unclear what _numbers mean. Conventionally, gene names should be in italics. "to a given AMPs" should be "AMP" singular?

We modified the figure legends as follows:

Figure 4 - Mutational profiles of 38 AMP-resistant lines. a) The figure shows cellular complexes and pathways mutated independently in multiple lines as a function of AMPs used during laboratory evolution. The color code indicates the number of individual mutations affecting a given cellular subsystem in lines evolved to a given AMP. Abbreviations; OM: outer membrane, PL: phospholipid, LPS: lipopolysaccharide, Δ : gene deletion. b) Heatmap shows mutation profile similarity of AMP-resistant lines. Mutation profile similarity between each pair of AMP-resistant lines was estimated by the Jaccard's coefficient between their set of mutated genes. Large deletions were counted as one gene. AMP-resistant lines are named after the AMP it has been exposed to during the evolutionary experiment, and the serial number of the corresponding population. For AMP abbreviations, see Supplementary Table 1.

Figure 5: Should the use of multiple tests not merit a correction for multiple comparisons, e.g. Bonferroni, which at the 'conventional' $\alpha = 0.05$ would result in a corrected α of $0.05/20 = 0.0025$. I'm also confused by the side-table and the bold part of the caption. I thought the caption suggests that lines carry a single mutation, but the table seems to show some strains with mutations in more than one gene?

Thank you for the suggestion. We used Dunnett's ANOVA post hoc test to identify strains with significant differences in the surface charge compared to the wild type strain. This test was used to compare each of the number of treatments with a single control. We modified the side-table and the figure legend, as requested:

Figure 5 – Surface charge measurement of AMP-adapted lines and the three reinserted mutants. Relative surface charge was measured as the relative change in the binding of the positively charged FITC-PLL compared to the wild type (see Methods). The table on the right side shows whether the adapted line carries a mutation in the *waa* or *mli* pathway genes or in the BasSR two-component system. AMP-resistant lines are named after the AMP it has been exposed to during the evolutionary experiment, and the serial number of the corresponding population. Boxplots show the median, first and third quartiles, with whiskers showing the 5th and 95th percentile. Significant differences compared to the wild type are marked with grey asterisks (* $P < 0.05$, ** $P < 0.01$, * $P < 0.001$, Dunnett's test).**

Figure 6: There are 3 'outliers' for antibiotics: trimethoprim, ampicilin and cefoxitin. I do not think it is appropriate to bin the antibiotics into a category and conduct a two-way test on AMPs vs Abs--- clearly there is heterogeneity in resistance to ABs conferred by overexpression. I think a nested

ANOVA design here would seem more appropriate (see e.g. <https://www.nature.com/articles/nmeth.3137>).

Thank you for raising this issue. Although we couldn't apply nested ANOVA due to large differences in the variance between drugs, we did perform a permutation test to address this issue. Instead of using pooled data, we calculated, for each drug, the mean log₁₀-transformed MIC values of the pooled gene overexpression library. Next, we used a permutation procedure to test whether the group of antibiotics shows a significantly higher MIC values than the group of AMPs. In the test, we randomly assigned drug treatments to sets of replicate MIC measurements on the pooled gene overexpression library and calculated the difference in mean MIC values between AMP and antibiotic treatments. This procedure was repeated 10000 times. Reassuringly, the real difference between the two groups was significantly higher than those in the permuted samples ($p < 0.0001$).

We modified the figure legend as follow:

Figure 6- Impact of gene amplification and foreign genes on resistance. a) Depicts the impact of gene overexpression on resistance level. They show the MIC provided by the ASKA plasmid library relative to the MIC of the wild-type carrying the empty ASKA plasmid (three biological replicates each). Altogether, the overexpression of the ASKA plasmid library resulted in significantly higher resistance to antibiotics (N=11) than for AMPs (N=14) ($P < 0.0001$, permutation test). Boxplots show the median, first and third quartiles, with whiskers showing the 5th and 95th percentile. **b)** Functional metagenomics of a soil library. Functional selection has revealed 41 distinct antibiotic resistance-conferring DNA contigs (red bars), while no AMP resistance-conferring contigs were identified ($P = 2 \times 10^{-16}$ from two-sided negative binomial regression). For AMP and antibiotic abbreviations, see Supplementary Tables 1-2.

Results:

L415: 'relatively immune to resistance' is vague, as it does not specify relative to what---to other AMPs? ABs? I wonder if a different word could be used instead of 'immune'---can something be 'relatively immune'?

We modified the text as follows:

We propose that these AMPs could serve as a promising base for the development of peptide based drug candidates with limited resistance.

Minor:

L356: remove the extra %

L358: associated to associated with?

L363: less fewer? (If referring to the #/proportion of amino acids)

L795: The definition of IPTG is given here, but comes after IPTG is first used in the text

L356 to L795: Done.

The usage of species names is non-standard in parts (i.e. full genus and species for first usage, and abbreviated genus afterward) but hopefully that can be addressed by typesetting. Similarly, in places both K12 and K-12 are used (I believe it's *Escherichia coli* str. K-12 substr. BW25113 'officially', for the others I'm not sure).

Throughout the manuscript could use checking that terms are used consistently.

Thank you, we check the terms.

Reviewer #3 (Remarks to the Author):

This manuscript from Spohn et al involves a study of the evolution of resistance to a diverse set of AMPs and antibiotics in *Escherichia coli*. Based on several different types of experimental data they suggest that AMPs are less prone to resistance development than classical small molecule antibiotics for four reasons: (i) in laboratory evolution experiments AMPs are less prone to resistance evolution than antibiotics, (ii) limited cross-resistance is seen between AMPs, (iii) increased copy number of native genes cannot confer AMP resistance and (iv) soil meta-genomic fragments cannot confer AMP resistance.

Overall I think this is a nice study with some important findings that are suitable for Nature Communications.

Thank you.

However, there are a few things the authors need to address before it would be acceptable.

1) The choice of soil metagenomics as a source for AMP-resistance genes is unclear to me. Are AMPs present in soil? If not, what was the reason to choose soil over other microbiomes that are more likely to encounter AMPs? Some more discussion on this would be helpful to follow the authors motivation for this particular experimental setup.

As AMPs are produced by a diverse set of organisms to promote immune defenses, nutrient acquisition or elimination of competitors from the environment, they are found in a many environments¹, including mammalian tissues, soil²⁻⁵ and aquatic environments. For example, *Bacillus polymyxa* inhabits soil, plant roots, and marine sediments³, and produces clinically relevant polymyxins. Soil is also an ancient reservoir of antibiotic resistance genes and it has been shown that antibiotic resistance genes can be exchanged between soil dwelling and pathogenic bacteria⁶. For all these reasons, we consider soil as a relevant source environment to study the mobilization potential of AMP-resistance genes.

1. Biswaro, L. S., da Costa Sousa, M. G., Rezende, T. M. B., Dias, S. C. & Franco, O. L. Antimicrobial Peptides and Nanotechnology, Recent Advances and Challenges. *Front. Microbiol.* 9, 855 (2018).
2. Baidara, P. *et al.* Characterization of two antimicrobial peptides produced by a halotolerant *Bacillus subtilis* strain SK.DU.4 isolated from a rhizosphere soil sample. *AMB Express* 3, 2 (2013).
3. Shaheen, M., Li, J., Ross, A. C., Vederas, J. C. & Jensen, S. E. *Paenibacillus polymyxa* PKB1 Produces Variants of Polymyxin B-Type Antibiotics. *Chem. Biol.* 18, 1640–1648 (2011).
4. Bizani, D. *et al.* Antibacterial activity of cerein 8A, a bacteriocin-like peptide produced by *Bacillus cereus*. *Int. Microbiol.* 8, 125–31 (2005).
5. Muhammad, N. *et al.* ISOLATION OPTIMIZATION AND CHARACTERIZATION OF ANTIMICROBIAL PEPTIDE PRODUCING BACTERIA FROM SOIL. *J. Anim. Plant Sci* 25,
6. Forsberg, K. J. *et al.* The Shared Antibiotic Resistome of Soil Bacteria and Human Pathogens. *Science (80-.)*. 337, 1107–1111 (2012).

2) A table should be provided that give the starting MIC values for the all species/drug combinations that have been tested. This data should already be available to the authors since the MIC's must have been determined to define the starting concentrations for the evolution experiment.

We agree that this information is necessary, see Supplementary table 14.

3) Why were some strains grown at 30°C and others at 37°C?

This was done for technical reasons and to ensure direct comparison of our results to previous works on similar subject (e.g. Lazar et al. Nature Communications 2014). Reassuringly, the MICs are insensitive to minor changes in temperature setting, see:

	MIC in 30 °C			MIC in 37 °C		
incubation time (hour)	24	48	72	24	48	72
TP11 (µg/mL)	5	5	6	5	6	6
PXB (µg/mL)	4.2	4.2	4.2	4.2	4.2	4.2

4) One major claim of the study is that resistance is not easily selected in vitro during an evolution experiment with increasing AMP-concentration. The authors should provide more information regarding the specific conditions of the cycling, for example how many cells were transferred during each cycling, how many generations of growth were achieved per cycle and what is the final cell density in the described MS-media. Using this information, it would also be very interesting to model what the minimal rate of emergence of potential rare resistance mutation should be to be selected under these experimental conditions.

Bell and Maclean (Trends in Microbiology 2018) provided a simple calculation for the likelihood that resistance to a novel agent will spread in laboratory evolution settings. They suggest that the power of experiments to detect resistance is determined by the number of mutations at each site in the genome. They argue that for a serial passage experiment, this number is approximately $Nt * m * g * r$, where Nt is the number of cells that are transferred to fresh media in each cycle, r is the number of replicate populations, g is the number of generations that elapsed during the experiment, and M is the mutation rate per nucleotide per replication.

In our case, these numbers are $N_t = 1.125 \cdot 10^7$ cells, $r = 10$, $g = 80-100$ generations (4-5 generations per transfer), $M (E. coli) = 2 \cdot 10^{-10}$ (Bell and Maclean, Trends in Microbiology 2018). $M = 2 \cdot 10^{-5} - 2 \cdot 10^{-6}$ in the case of *E. coli* mutD5 mutator strain (Schaaper PNAS 1988) employed in TP11 selection.

5) Cells were grown for 72h each cycle. Are active AMP concentrations constant during this time? It is known that some AMP have limited stability under laboratory conditions and concentrations might be reduced during incubation, for example, by absorption to plastic). Thus, could it be possible that the active concentration decreases which subsequently prevents enrichment of mutants? This might be especially important when the pharmacodynamics response curve is very steep and the selective window is narrow (see also point 6 below). An easy way to address this would be by two separate microdilution MIC determinations, one with freshly prepared media, and one with media (+AMP) that was pre-incubated for 72h. Ideally, the MIC should be identical for both assays.

We performed the experiments as requested on selected AMPs, many of which with limited resistance. We found that neither pre-incubation (72 hours) nor the exact timing of MIC measurement had any major effect on the MICs:

Time allowed for bacterial growth (hours)	MIC (freshly prepared AMP)			MIC (pre-incubated AMP)		
	24	48	72	24	48	72
TP11 (µg/mL)	5	5	5	6	6	6
PXB (µg/mL)	4.2	4.2	4.2	5	5	5
LL37 (µg/mL)	14.5	17.4	17.4	17.4	17.4	17.4
BAC5 (µg/mL)	14.5	14.5	14.5	14.5	14.5	14.5

6) I am glad to see that the authors discuss the paper (ref 53) regarding the steeper pharmacodynamics response and narrower selective window of AMPs compared to antibiotics. I suspect that this is a main factor that reduces the risk of AMP resistance development in clinical settings. Perhaps the author could in the discussion, in addition to the 4 reasons they list in the Abstract for why AMP resistance evolution is slower compared to antibiotics, explicitly include the PD

aspects as a 5th reason. Also, it would have been wonderful if the authors had included such PD data for the AMPs and antibiotics studied (however it is not something necessarily needed for this paper).

We briefly write about this possibility in the discussion:

Sum, our analysis indicates that the frequency by which AMP resistance arises is relatively low and even if they occur, such genetic changes provide only relatively low levels of resistance. Why is it so? We hypothesize two complementary mechanisms that should be studied in future works. First, evolution of AMP resistance may be hindered by the limited availability of resistance mutations..... Second, the AMP dose range under which resistance can evolve may be limited. Indeed, AMPs significantly differ from antibiotics in their pharmacodynamics properties. In particular, their bactericidal activity is generally characterized by a sharp increase within a narrow dose range, which is in contrast with the pharmacodynamics of many antibiotics⁵⁷. ... “

7) The authors show that the fitness costs associated with AMP resistance mutations is lower than that associated with antibiotic resistance mutations. However, I am not sure if one can draw the conclusion from this that resistance mutations for the specific AMPs are or are not observed because of being costly/detrimental.

There may be a misunderstanding here. We do not claim that shortage of resistance against certain AMPs would reflect of being detrimental for the host bacteria.

The analysis, as I understand it, does not include the mutants observed for CP1, PGLA, IND and PLEU, which are among the AMPs against which development of resistance is limited. I understand that the authors are trying to extrapolate by making fitness cost a general phenotype, but I am not sure one can do that. Also, the authors should also mention that these costs are on isolated mutants and not on reconstructed mutants, and might thus have other mutations present as well.

In the result section, it is now explicit that we study lines isolated from the final day of laboratory evolution:

In this analysis, we focused on 60 antibiotic-adapted, 38 AMP-adapted strains, all of which displayed at least 2-fold increment in resistance level to the drug they had been exposed to during the course of laboratory evolution. ... In total, 96.7% of the antibiotic-resistant lines showed

significantly reduced growth compared to the wild-type strain, while this figure is 92.1% in the case of AMP-adapted lines.

We note that none of the strains displayed significantly improved fitness (growth rate) in antibiotic free medium compared to the starting wild-type. Therefore, accumulation of adaptive mutations unrelated to drug resistance is expected to be rare during the course of laboratory evolution.

8) The authors write "...Reassuringly, 94% of all point mutations were non-synonymous, indicating that their accumulation was driven by positive selection." I am not sure if accumulation of non-synonymous mutations over only 120 generations can be used to make this argument, especially because one generally expects to observe many more non-synonymous mutations in such experiments.

Thank you for the observation. We modified the text as follows:

To statistically test whether the ratio of non-synonymous to synonymous SNPs was higher than expected based on a neutral model of evolution, we employed an established method (Barrick et al. Nature 2009), that is especially well-suited for experimental evolution studies with limited number of observed mutations (Szamecz et al. Plos Biology 2014). Briefly, we took all different point mutations observed in protein coding regions and calculated the probability that 94% or more substitutions would result in a non-synonymous substitution if it occurred in a random coding position. The excess of non-synonymous substitution observed in the evolved genomes was significant ($p = 0.000004$).

9) The type of reconstructed mutations in BasS, WaaY and mlaD should be specified, especially if there is any knowledge of whether these are loss of function mutations are not.

Done.

10) The authors should discuss more about the MICs of CPI and TPII for Enterobacter cloacae (Supplementary table 11), since they are much higher than for the rest of the Gram-negatives tested.

We failed to find any relevant information that could satisfactorily explain this pattern.

Reviewers' Comments:

Reviewer #1:

Remarks to the Author:

Dear editor,

I believe this is an important contribution to the field. I am happy with the authors response which are clear and relevant. My only request is that the discussion about the controversy around the toxicity/activity of these peptides come out clearer in the discussion. I think this is one of the more important conclusions of the work and obviously one of the "important motivations" for the study. Why not just add a few sentences in the line of the response of the reviewer in this regard and the references. Efforts in developing peptides into new antimicrobials are frequently hampered by difficulties in separating toxicity and activity. The index and the existences of peptides with high activity and low toxicity could reignite the general interest in antimicrobial peptides as potential novel antimicrobials.

Reviewer #2:

Remarks to the Author:

I am satisfied the authors have adequately addressed my comments from the first round of review. As stated in my original comments, this paper presents novel results using two interesting methods (plasmid over-expression and environmental DNA). This version makes fewer unjustified (and unnecessary, in my opinion) claims about clinical relevance. The statistics in this revised version are better justified. I believe this paper will be interesting to evolutionary biologists and those working on antimicrobial resistance.

- Danna Gifford

Reviewer #3:

Remarks to the Author:

Thank you for responding to my concerns in a satisfactory way.

/Dan Andersson

Response to the reviewer

Reviewer #1 (Remarks to the Author):

Dear editor,

I believe this is an important contribution to the field. I am happy with the authors response which are clear and relevant. My only request is that the discussion about the controversy around the toxicity/activity of these peptides come out clearer in the discussion. I think this is one of the more important conclusions of the work and obviously one of the "important motivations" for the study. Why not just add a few sentences in the line of the response of the reviewer in this regard and the references. Efforts in developing peptides into new antimicrobials are frequently hampered by difficulties in separating toxicity and activity. The index and the existences of peptides with high activity and low toxicity could reignite the general interest in antimicrobial peptides as potential novel antimicrobials.

We agree and added the following sentences to the discussion:

“We note however, that existing data on the toxicity of the tachyplesin antimicrobial family are controversial ^{59,60,61,49}. Promisingly, and in agreement with our results, a previous work showed that CP1 has negligible toxicity or haemolytic activity ⁶².”

Reviewer #2 (Remarks to the Author):

I am satisfied the authors have adequately addressed my comments from the first round of review. As stated in my original comments, this paper presents novel results using two interesting methods (plasmid over-expression and environmental DNA). This version makes fewer unjustified (and unnecessary, in my opinion) claims about clinical relevance. The statistics in this revised version are better justified. I believe this paper will be interesting to evolutionary biologists and those working on antimicrobial resistance.

Reviewer #3 (Remarks to the Author):

Thank you for responding to my concerns in a satisfactory way.

Response to the editor

MAIN TEXT

* Please use the present tense when discussing the current work in the Introduction.

We have re-written that part of the introduction.

* Please shorten all subheadings in the Results section to fewer than 60 characters including spaces.

The subheadings are modified.

LANGUAGE AND STYLE

* Please make sure that mathematical terms throughout your manuscript and Supplementary Information (including in figures, figure axes, and legends) conform strictly to the following guidelines. Equations should be supplied in editable format, and not as images. Scalar variables (e.g. x , V , χ) should be typeset in italic, whereas multi-letter variables should be formatted in roman. Constants (e.g. \hbar , G , c) should be typeset in italics (the only exceptions being e , i , π , which should be typeset in Roman) and vectors (such as r , the wavevector k , or the magnetic field vector B) should be typeset in bold without italics. In contrast, subscripts and superscripts should only be italicised if they too are variables or constants. Those that are labels (such as the 'c' in the critical temperature, T_c , the 'F' in the Fermi energy, E_F , or the 'crit' in the critical current, I_{crit}) should be typeset in roman. To avoid doubt, unit dimensions should be expressed using negative integers (e.g. $\text{kg m}^{-1} \text{s}^{-2}$, not kg/ms^2) or the word 'per'.

We modified the unit dimensions.

* Wherever p-values are stated in the text and figure legends, please also state the name of the statistical test.

We complemented the missing information.

METHODS AND DATA

* We allow only one level of subheadings in the Methods section. Please remove secondary subheadings.

We removed secondary subheadings.

* In the Methods section, please provide sufficient information such that the experiments could reasonably be reproduced without reference to other papers, and avoid use of the term "as described previously".

We modified the text if it was necessary.

* All Nature Communications manuscripts must include a section titled "Data Availability" as a separate section after the Methods section and before the References. For more information on this policy, and a list of examples, please see <http://www.nature.com/authors/policies/data/data-availability-statements-data-citations.pdf>

We included the Data Availability section.

* In an effort to ensure reproducibility of research data, we now also require that you provide a separate Source Data file. The source data file should, as a minimum, contain the raw data underlying all reported averages in graphs and charts, and uncropped versions of any gels or blots presented in the figures. To learn more about our motivation behind this policy, please see <https://www.nature.com/articles/s41467-018-06012-8>.

Within the source data file, each figure or table (in the main manuscript and in the Supplementary Information) containing relevant data should be represented by a single sheet in an Excel document, or a single .txt file or other file type in a zipped folder. Blot and gel images should be pasted in and labelled with the relevant panel and identifying information such as the antibody used. We also encourage you to include any other types of raw data that may be appropriate. An example source data file is available demonstrating the correct format: <https://www.nature.com/documents/ncomms-example-source-data.xlsx>

The file should be labelled 'Source Data', with the title and a brief description included in your cover letter, and should be mentioned in all relevant figure legends using the template text below: "Source data are provided as a Source Data file."

We upload the Source Data file and mentioned in all figure legends.

* Please ensure that all novel nucleotide sequences are deposited in the NCBI Genbank nucleotide database, and that accession codes are provided in the "Data Availability" section.

We upload the nucleotide sequences and provided in the "Data Availability" section.

* To ensure correct hyperlinking of the accession codes in your manuscript, please add the hyperlink or DOI in square brackets directly after the code throughout (for example, "5XRN [<http://dx.doi.org/10.2210/pdb5XRN/pdb>]", "1483958 [<https://dx.doi.org/10.5517/ccdc.csd.cc1t5m6>]", "SRP109982 [<https://www.ncbi.nlm.nih.gov/sra/?term=SRP109982>]" or "NQLW00000000 [https://www.ncbi.nlm.nih.gov/assembly/GCA_002312845.1/]").

Done

* Please add a reference to the source data file in the "Data Availability" section. For example: "The source data underlying Figs 1a, 2a–d, 6d, h and 7c and Supplementary Figs 1a and 5d are provided as a Source Data file."

Done

END NOTES

* Please ensure that all in-text citations to references (e.g. "Smith et al. show...") are followed by their corresponding reference citation number from the reference list, and that the references are numbered in the order they appear in the text (then tables and figures).

Done

* Only papers that have been published or accepted by a named publication or recognised preprint server should be in the numbered list.

Done

DISPLAY ITEMS

* Please check whether your manuscript or Supplementary Information contain third-party images, such as figures from the literature, stock photos, clip art or commercial satellite and map data. We strongly discourage the use or adaptation of previously published images, but if this is unavoidable, please request the necessary rights documentation to re-use such material from the relevant copyright holders and return this to us when you submit your revised manuscript.

We do not use third-party images.

* Please ensure that figure legend titles are brief - they should not occupy more than one line in the final proof.

We modified the figure legends accordingly.

* Where p-values are presented as symbols/letters, please ensure that these are defined in the relevant figure legend, and the statistical test used to generate them is stated.

These are defined in the figure legend.

* Please ensure that all colour scales are defined in either the figure or its associated legend.

Figures or figure legends contain this information.

* Please ensure that all elements of boxplots are defined in the associated figure legend (centre line, bounds of box and whiskers).

Figure legends contain this information.

* Some figures in your paper include bar charts. Please overlay the corresponding data points (as dot plots) in the bar charts.

In case of figure 6b the bars represent quantities resulting from one screen. In other figures (Figure 1A, 1B, 5, 8A) we show the unique data points, if there were less than or equal to 10 data points.

* Please define any new abbreviations, symbols or colours present in your figures in the associated legends. Please do not use symbols in your legend, instead please write out the symbols in words (blue circles, red dashed line, etc.).

Done

* In each figure and supplementary figure where error bars are used, they must be defined. One statement at the end of each figure is sufficient if the error bars are equivalent throughout the figure.

Done

SUPPLEMENTARY INFORMATION

* We do not typeset Supplementary Information files; they will be uploaded with the published article as they are submitted with the final version of your manuscript. Any tracked changes should be removed from the file and the file should be provided as a PDF file. Supplementary Figures do not need to be provided separately.

Done and uploaded.

* Supplementary References should appear at the end of the Supplementary Information file, and should be self-contained and numbered from 1. References mentioned in both the main text and the Supplementary Information should be part of both reference lists so that the Supplementary Information does not refer to the reference list in the main paper and vice versa.

We moved the reference list to the end of Supplementary information.

* Please supply legends for each Supplementary Movie/Audio/Data file in your cover letter (not in the Supplementary Information file). Please label each files as Supplementary Movie/Audio/Data 1, etc.

We modified and re-numbered the Supplementary Data files.

* Large datasets should be supplied as Supplementary Data files, whereas smaller tables should be supplied as supplementary tables.

Done

* Within the Source Data file, each figure or table (in the main manuscript and in the Supplementary Information) containing relevant data should be represented by a single sheet in an Excel document, or a single .txt file or other file type in a zipped folder. Blot and gel images should be pasted in and labelled with the relevant panel and identifying information such as the antibody used. We also encourage you to include any other types of raw data that may be appropriate. An example Source Data file is available demonstrating the correct format: <https://www.nature.com/documents/ncomms-example-source-data.xlsx> The file should be labelled "Source Data", with the title and a brief description included in your cover letter, and should be mentioned in all relevant figure legends using the template text below: "Source data are provided as a Source Data file."

We performed Source Data file accordingly.

* We encourage increased transparency in peer review by publishing the reviewer comments and author rebuttal letters of our research articles, if the authors agree. Such peer review material is made available as a supplementary peer review file. Please state in the cover letter 'I wish to participate in transparent peer review' if you want to opt in, or 'I do not wish to participate in transparent peer review' if you don't. Failure to state your preference will result in delays in accepting your paper for publication. Please note: we allow redactions to authors' rebuttal and reviewer comments in the interest of confidentiality. If you are concerned about the release of confidential data, please let us know specifically what information you would like to have removed. Please note that we cannot incorporate redactions for any other reasons. Reviewer names will be published in the peer review files if the reviewer signed the comments to authors, or if reviewers explicitly agree to release their name. For more information, please refer to our FAQ page at: <https://www.nature.com/documents/ncomms-transparent-peer-review.pdf>

We uploaded a supplementary peer review file.

* An updated editorial policy checklist that verifies compliance with all required editorial policies must be completed and uploaded with the revised manuscript. All points on the policy checklist must be addressed; if needed, please revise your manuscript in response to these points. Please note that this form is a dynamic "smart pdf" and must therefore be downloaded and completed in Adobe Reader, instead of opening it in a web browser. Editorial policy checklist: <https://www.nature.com/authors/policies/Policy.pdf>

Editorial policy checklist is completed and uploaded.

* An updated reporting summary must be completed and uploaded with the revised manuscript. All points on the reporting summary must be addressed; if needed, please revise your manuscript in response to these points. Please note that this form is a dynamic "smart pdf" and must therefore be downloaded and completed in Adobe Reader, instead of opening it in a web browser. Reporting summary: <https://www.nature.com/authors/policies/ReportingSummary.pdf>

Reporting summary is completed and uploaded.

* Your paper will be accompanied by a two-sentence Editor's summary, of between 250-300 characters including spaces, when it is published on our homepage. Could you please approve the draft summary below or provide us with a suitably edited version.

"Antimicrobial peptides (AMPs) are emerging as drug candidates, but the risk of pathogen resistance is not well understood. Here, the authors investigate AMP resistance evolution in E. coli, finding physicochemical features that make AMPs less prone to resistance and no cross or horizontally-acquired resistance. "

We approved the draft.

* As part of our efforts to communicate our content to a wider audience, we endeavour to highlight papers published in Nature Communications on the journal's Twitter account (@NatureComms). If you would like us to mention authors, institutions or lab groups in these tweets, please provide the relevant twitter handles in your cover letter upon resubmission.

OK.

* If you opted into the journal hosting details of a preprint version of your manuscript via a link on our dedicated website (<https://nature-research-under-consideration.nature.com>), it will remain on this site while you are revising your manuscript, as we consider the file to remain active. Should you wish to remove these details, please email naturecommunications@nature.com indicating your manuscript number and the link on our website that was previously sent to you.

Please see our pre-publicity policy at

<http://www.nature.com/authors/policies/confidentiality.html>

For more information, please refer to our FAQ page at <https://nature-research-under-consideration.nature.com/posts/19641> frequently-asked-questions

We do not have pre-print version.